# SKP2 attenuates autophagy through Beclin1-ubiquitination and its inhibition reduces MERS-Coronavirus infection

Nils C. Gassen[1,2]*, Daniela Niemeyer[3,4], Doreen Muth[3,4], Victor M. Corman[3,4], Silvia Martinelli[1], Alwine Gassen[5], Kathrin Hafner[1], Jan Papies[3,4], Kirstin Mösbauer[3,4], Andreas Zellner[1], Anthony S. Zannas[1,6,7,8], Alexander Herrmann[9], Florian Holsboer[1], Ruth Brack-Werner[9], Michael Boshart[5], Bertram Müller-Myhsok[1,10,11], Christian Drosten[3,4], Marcel A. Müller[3,4,12,14] & Theo Rein[1,13,14]*

Autophagy is an essential cellular process affecting virus infections and other diseases and Beclin1 (BECN) is one of its key regulators. Here, we identified S-phase kinase-associated protein 2 (SKP2) as E3 ligase that executes lysine-48-linked poly-ubiquitination of BECN1, thus promoting its proteasomal degradation. SKP2 activity is regulated by phosphorylation in a hetero-complex involving FKBP51, PHLPP, AKT1, and BECN1. Genetic or pharmacological inhibition of SKP2 decreases BECN1 ubiquitination, decreases BECN1 degradation and enhances autophagic flux. Middle East respiratory syndrome coronavirus (MERS-CoV) multiplication results in reduced BECN1 levels and blocks the fusion of autophagosomes and lysosomes. Inhibitors of SKP2 not only enhance autophagy but also reduce the replication of MERS-CoV up to 28,000-fold. The SKP2-BECN1 link constitutes a promising target for host-directed antiviral drugs and possibly other autophagy-sensitive conditions.

[1] Department of Translational Research in Psychiatry, Max Planck Institute of Psychiatry, Kraepelinstr. 10, 80804 Munich, Germany. [2] Department of Psychiatry and Psychotherapy, University of Bonn, Venusberg Campus 1, 53127 Bonn, Germany. [3] Institute of Virology, Charité-Universitätsmedizin Berlin, Humboldt-Universität zu Berlin, and Berlin Institute of Health, Charitéplatz 1, 10117 Berlin, Germany. [4] German Centre for Infection Research (DZIF), Berlin, Germany. [5] Faculty of Biology, Genetics, Ludwig-Maximilian-University Munich (LMU), 82152 Martinsried, Germany. [6] Department of Psychiatry and Behavioral Sciences, Duke University Medical Center, Durham, NC 27710, USA. [7] Department of Psychiatry, University of North Carolina at Chapel Hill, 438 Taylor Hall, 109 Mason Farm Road, Chapel Hill 27599-7096 NC, USA. [8] Department of Genetics, University of North Carolina at Chapel Hil, Chapel Hill 27599 NC, USA. [9] HIV-Cell-Interactions Group, Institute of Virology, German Research Center for Environmental Health, Ingolstädter Landstr. 1, 85764 Neuherberg, Germany. [10] Institute of Translational Medicine, University of Liverpool, L69 3BX Liverpool, UK. [11] Munich Cluster for Systems Neurology - SYNERGY, Feodor-Lynen-Str. 17, 81377 Munich, Germany. [12] Martsinovsky Institute of Medical Parasitology, Tropical and Vector Borne Diseases, Sechenov University, 2-4 Bolshaya Pirogovskaya st., 119991 Moscow, Russia. [13] Faculty of Medicine, Physiological Chemistry, Ludwig-Maximilian-University Munich (LMU), 82152 Martinsried, Germany. [14]These authors jointly supervised this work: Marcel A. Müller, Theo Rein. *email: ncgassen@psych.mpg.de; theorein@psych.mpg.de

Macroautophagy (subsequently referred to as autophagy) is a conserved cytosolic degradation process in eukaryotes that maintains macromolecule, energy and organelle homeostasis[1]. It requires a fine-tuned sequence of molecular actions largely executed by autophagy-related genes (ATGs). In the initial steps, a phagophore consisting of a double-membrane sheet is nucleated, amplified and finally closed to form a vesicular structure called an autophagosome that engulfs cellular material for degradation[1]. Degradation is accomplished after the fusion of autophagosomes (AP) with lysosomes to form autolysosomes (AL).

A more recently emerged regulator of autophagy is the HSP90 cochaperone FK506 binding protein (FKBP)51[2]. In general, FKBP51 is a stress-regulated protein involved in numerous pathways, likely through scaffolding regulatory protein interactions[3]. FKBP51-induced autophagy is accompanied by elevated levels of Beclin1 (BECN1) through a yet unknown mechanism. This multifunctional protein is crucial in autophagy and membrane trafficking processes[4]. In mammals, BECN1 is an essential component of two phosphatidylinositol 3-kinase (PI3K) complexes through its interaction with either Barkor (BECN1-associated autophagy-related key regulator)/ATG14 or UVRAG (UV radiation resistance-associated gene)[5]. The ATG14-complex is important for the initial nucleation steps of autophagy, whereas the UVRAG containing complex functions in endocytic trafficking and autophagosomal maturation. ATG14 is also found on mature autophagosomes where it enhances tethering and fusion of autophagosomes with lysosomes[6].

The regulation of BECN1 through post-translational modification is well documented, and in particular involves phosphorylation and ubiquitination[4]. The oncogenic kinase AKT1 phosphorylates BECN1 at positions S234, S295, which leads to sequestration of this peripheral membrane binding protein[7] to the cytoskeleton with the result of inhibition of autophagy[8]. Several other autophagy-regulating kinases (both stimulatory and inhibitory) also target BECN1[4]. Furthermore, distinct E3 ligases of different types modify BECN1 with K11-, K48- or K63-linked poly-ubiquitins[9], complemented by ubiquitin-specific peptidases (USPs)[10,11]. Depending on the site and type of linkage of BECN1 ubiquitination, autophagy is inhibited or enhanced; the underlying mechanisms involve degradation of BECN1 and modulation of its protein interactions[9,12].

Studies investigating the interplay between autophagy and viruses revealed that some viruses use parts of the autophagy machinery for their own replication but also have evolved strategies to escape autophagic degradation. Congruently, autophagy-inducing agents can have antiviral effects[13]. In case of coronaviruses (CoV), the formation of replication complexes at double membrane vesicles (DMV) and, most likely, of infectious particles depend on ER-derived membranes, as does autophagy[14–16]. Thus, current knowledge suggests that CoV interact differentially with components of the autophagic pathway with the potential for both the utilization of autophagy components for virus replication and for the attenuation of autophagy[17]. Substances inhibiting the generation of DMVs have been shown to be broadly reactive against CoV replication in vitro[18]. Importantly, this included CoV of different genera including the emerging Middle East respiratory syndrome (MERS)-CoV, a paradigmatic emerging virus that entails considerable pandemic risks due to its prevalence in dromedary populations in Africa and the Arabian Pensinsula and its transmission via the respiratory tract causing a severe form of pneumonia in humans[18–21]. Since its discovery in 2012, the WHO has reported 2468 human MERS-CoV infections in 27 countries, of which at least 851 were fatal (http://www.who.int/emergencies/mers-cov/en/).

Currently, there is no clinically-approved treatment for MERS-CoV infection. While holding promise as a broad-range host-directed antiviral target, in general[13], the role of autophagy in the MERS-CoV life cycle, if any, is unknown. Nevertheless, a kinome analysis of MERS-CoV-infected cells detected phosphorylation changes of several regulatory kinases including AKT1 and mTOR that have been linked to autophagy in other studies[22,23]. Although pharmacological interventions of broadly active AKT1- and mTOR-pathway inhibitors showed promise for antiviral therapy against several viruses including MERS-CoV[22], the identification of downstream molecules may facilitate a more targeted therapeutic approach with putative reduced side effects in patients.

A role of autophagy is suggested for another CoV, the mouse hepatitis virus (MHV), by the observation that multiplication of, and immune evasion by the virus is highly dependent on the formation of convoluted membranes and DMVs that are reminiscent of autophagosomes[24]. However, the link of MHV replication to these DMVs did not appear to involve autophagy as the deletion of the pivotal autophagy genes ATG7 or ATG5 did not affect MHV replication[25,26]. Of note, also the induction of autophagy by starvation did not significantly change MHV replication[26]. On the other hand, results of an earlier study employing ATG5 knockout cells suggested that autophagy is required for the formation of DMV-bound MHV replication complexes thereby significantly enhancing the efficiency of viral replication[16]. Furthermore, pharmacological or genetic manipulation of autophagy showed that replication of another CoV, the Transmissible Gastroenteritis virus (TGEV), is negatively regulated by autophagy[27]. In contrast, another study reported enhancement of TGEV replication by autophagy[28]. Thus, no general role of autophagy in CoV replication could be established yet.

Here, we aim to elucidate the mechanisms controlling BECN1 protein levels. We find that S-phase kinase-associated protein 2 (SKP2) executes lysine-48-linked poly-ubiquitination of BECN1; its activity is regulated through phosphorylation under the control of FKBP51 involving AKT1 and PHLPP. Small molecule inhibitors of SKP2 enhance autophagy and reduce replication of MERS-CoV, pointing to the prospect of their therapeutic usefulness.

## Results

**FKBP51 increases BECN1 stability.** In search for a mechanism of the previously reported increase of the pivotal autophagy regulator BECN1 driven by FKBP51[2] we considered effects on mRNA and protein level. In direct comparison to the highly homologous FKBP52, a known counter-player of FKBP51[29], only FKBP51 increased BECN1 levels upon ectopic expression[3] (Fig. 1a). Regulation of BECN1 protein stability through the ubiquitin-proteasome system was indicated by using the proteasome inhibitor MG132, which increased the levels of BECN1 and the extent of its ubiquitination (Fig. 1b, Supplementary Fig. 1a). The use of ammonium chloride to inhibit lysosome-mediated proteolysis confirmed proteasomal degradation of BECN1 (Supplementary Fig. 1b). Ectopic expression of FKBP51 was similarly efficient in stabilising BECN1 as proteasome inhibition by MG132 (Fig. 1c, d). A protein degradation assay based on a pulse-chase using Halo-tagged BECN1[30] confirmed that FKBP51 stabilises BECN1 (Fig. 1e, f). These results also revealed a high turn-over rate of BECN1 ($t_{1/2} < 1$ h).

**FKBP51 associates with SKP2 regulating BECN1 stability.** FKBP51 recently emerged as a scaffolder protein organising various regulatory protein complexes[2,31–33]. We thus considered an FKBP51-controlled association of BECN1 with E3 ubiquitin ligases or USPs. Based on the results of an FKBP51 mammalian-2-hybrid interaction screen[34], we first verified the association of

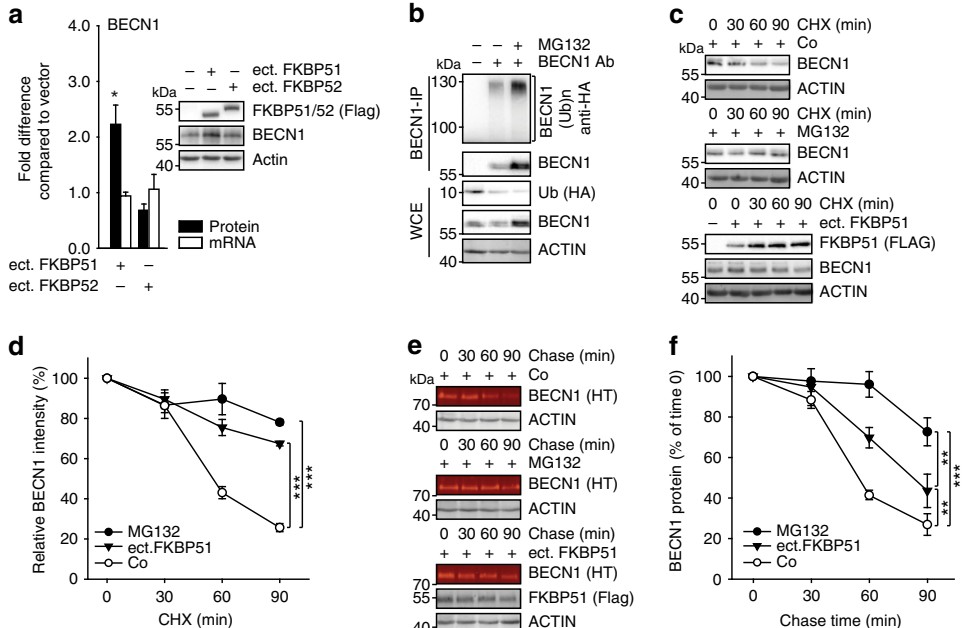

**Fig. 1 FKBP51 increases BECN1 stability. a** FKBP51 increases BECN1 protein levels. HEK293 cells were transfected with either Flag-tagged FKBP51 or FKBP52 expressing plasmid or control vector. BECN1 mRNA and protein were determined after 72 h. **b** BECN1 is subject to proteasomal degradation. HEK293 cells were transfected with Ubiquitin-HA expressing plasmid and treated with the proteasome inhibitor MG132 (10 μM, 2 h) as indicated. BECN1 was immunoprecipitated from whole cell extracts and probed for ubiquitination by western blotting (quantification Supplementary Fig. 1a). **c–f** FKBP51 delays degradation of BECN1. HEK293 cells were transfected with vector control (Co) or FKBP51 expressing plasmid for 72 h and treated with the translation inhibitor cycloheximide (CHX, 30 μg mL$^{-1}$) as indicated in **c**, **d**. In **e**, **f**, HEK293 cells were transfected with Halo-tagged BECN1 expressing plasmid together with FKBP51-Flag expressing plasmid or vector, cultivated with halogenated dye (100 nM, R110Direct, pulse) overnight, switched to medium without dye (chase) and treated with MG132 or vehicle for the indicated times. In all panels, error bars denote the standard error of the mean, derived from n = 3 biologically independent experiments. *p < 0.05, **p < 0.01, ***p < 0.001 (one-way ANOVA in **a**, two-way ANOVA in **d**, **f** details in Supplementary Table 1). Source data and blot collections are provided as a Source Data file.

FKBP51 with the USPs 18 and 36, and with the E3 ligase SKP2 by immunoprecipitation in HEK293 cells (Fig. 2a). TRAF6 that is known as an E3 ligase of BECN1[35] and interactor of FKBP51[36] served as a control. Of these FKBP51 interactors, SKP2, USP18 and USP36 have the potential to influence K48-linked poly-ubiquitination of BECN1. Co-immunoprecipitations of BECN1 in HEK293 cells revealed association with SKP2 only (Fig. 2b), which has not been described as an E3 ligase of BECN1 before. However, inhibition of SKP2 has been previously linked to induction of autophagy[37,38].

Since we aimed at understanding the mechanism of FKBP51-directed increase in BECN1, we focussed in the following on proteins we found associated with both FKBP51 and BECN1. To assess the effect of SKP2 on BECN1 stability, we monitored the protein levels of BECN1 after the addition of the translation inhibitor cycloheximide (CHX) in the presence of over-expressed SKP2 or TRAF6 (again used as control), respectively. SKP2 affected BECN1 stability, leading to faster proteasomal degradation, while the K63-E3 ligase TRAF6 had no effect (Fig. 2c, d). Similar results were obtained in a pulse-chase experiment showing that SKP2 reduced BECN1 by about 20% compared to the vector control already at 30 min (Fig. 2e, f).

Further corroborating the function of SKP2 on BECN1, siRNA directed against Skp2 produced increased levels of BECN1 and the SKP2 target P27 along with enhanced lipidation of LC3B (typically detected as the ratio LC3B-II/I) (Fig. 2g, Supplementary Fig. 1c). Ectopic expression of SKP2 exerted the opposite effect (Fig. 2h, Supplementary Fig. 1d). Decreasing SKP2 by siRNA prevented BECN1 ubiquitination (Fig. 2i). In addition, stability assays using either CHX or the Halo-tagged BECN1 further established that reducing SKP2 levels with siRNA increases

BECN1 stability leading to almost 3-fold higher levels at 90 min (Fig. 2j–m).

**Phosphorylation-controlled ubiquitination of BECN1 by SKP2.** Next, we examined whether SKP2 ubiquitinates BECN1. Ubiquitination frequently forms chains linking several ubiquitin units to distinct lysine (K) residues. The type of lysine linkage determines protein fate and function[39]. Using a set of ubiquitin mutants, we found that SKP2 leads to K48-linked poly-ubiquitination (Fig. 3a), which marks proteins for proteasomal degradation[39]. Using a freely available mass-spectrometry database that proposes ubiquitination sites in mammalian proteins[40], we mutated various lysines in BECN1 and identified K402 in BECN1′s ECD (evolutionary conserved domain) as the attachment site of SKP2-directed ubiquitination (Fig. 3b, Supplementary Fig. 1e). The potential recruitment of SKP2 by FKBP51 raises the question how this may lead to reduced rather than increased ubiquitination of BECN1. Since FKBP51 inactivates the kinase AKT1 through complex formation with the phosphatase PHLPP[2,31], and AKT1 regulates SKP2 through phosphorylation at S72[41], we used ectopic expression of FKBP51-Flag and co-immunoprecipitation to probe for protein interactions and the phosphorylation status of SKP2. The association between FKBP51 and BECN1, AKT1, PHLPP, and SKP2 was accompanied by a decreased phosphorylation of AKT1 at position S473 and of SKP2 at position S72 (Fig. 3c–e).

These results suggest a mechanism where FKBP51 enacts a cascade of activity changes in AKT1 and SKP2 by the recruitment of PHLPP (Fig. 4a). We followed several approaches to substantiate the functionality of this cascade, first by employing SKP2 mutants that either mimic phosphorylation (S72D/S75D)

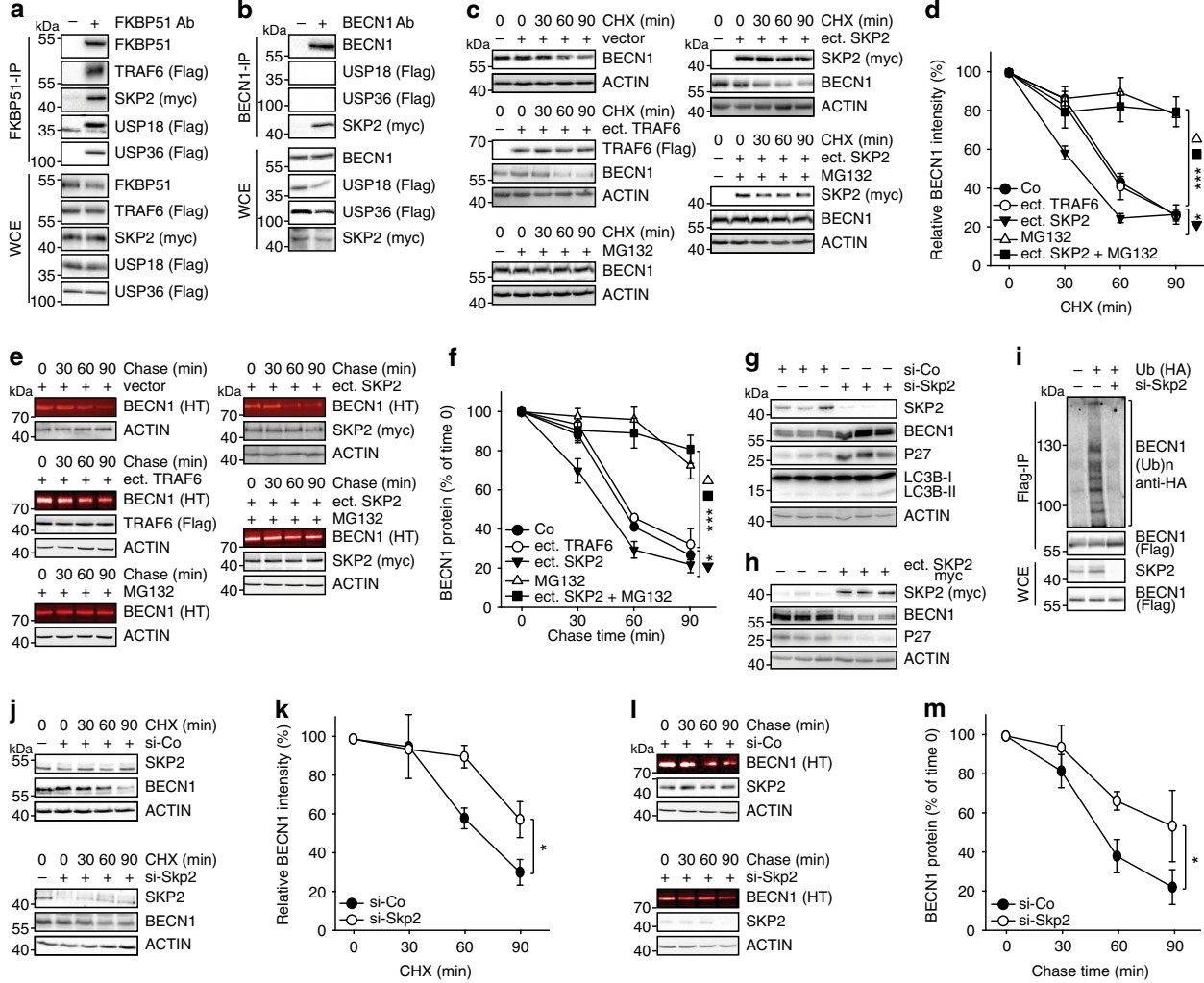

**Fig. 2 FKBP51 associates with SKP2 that regulates BECN1 stability. a** HEK293 cells were transfected with plasmids expressing the indicated E3 ligases and USPs; after 72 h, cells were either harvested for immunoprecipitation of FKBP51. **b** BECN1 associates with SKP2, but not with USP18 or USP36. HEK293 cells were transfected with vectors expressing tagged SKP2, USP18 and USP36; after 72 h, cells were harvested for immune-precipitation with either control IgG or anti BECN1 antibody. **c**, **d** SKP2 affects BECN1 stability. HEK293 cells were transfected with plasmids expressing SKP2 or TRAF6 and exposed to MG132 (10 μM, 2 h) where indicated and to cycloheximide (CHX, 30 μg mL$^{-1}$) after 72 h for the durations indicated to monitor the decay of BECN1. **e**, **f** The conditions of c,d were also used in the pulse-chase assay, performed as in Fig. 1e, f, to determine BECN1 stability. **g**, **h** Cellular SKP2 levels determine BECN1 levels. SKP2 was either down-regulated by using siRNA (**g**) or up-regulated by transient transfection (**h**) and the levels of the indicated proteins were measured by western blotting (P27 served as control). Quantifications for g and h are provided in Supplementary Fig. 1c, d. **i** Knock-down of SKP2 by siRNA decreases BECN1 ubiquitination (assay as in Fig. 1b). **j**–**m** Stability assays were performed like in **c**–**f** to determine the effect of SKP2-targeting siRNA on BECN1 stability. WCE = whole cell extract. In all panels, error bars denote the standard error of the mean, derived from $n = 3$ biologically independent experiments. *$p < 0.05$, **$p < 0.01$, ***$p < 0.001$ (two-way ANOVAs, details in Supplementary Table 1). Source data and blot collections are provided as a Source Data file.

or cannot be phosphorylated (S72A)[41]. The phosphomimetic mutant still decreased BECN1 stability, but the S72A mutant showed no effect (Fig. 4b, c), consistent with the proposed cascade (Fig. 4a). We further made use of a well-established PHLPP inhibitor[42] that led to enhanced phosphorylation of AKT1 and SKP2 (Fig. 4d, Supplementary Fig. 1f). A pulse-chase assay showed that the stability of BECN1 decreased by using the PHLPP inhibitor (Fig. 4e, f). This inhibitor had no effect on the stability of the ubiquitination-protected K402R mutant of BECN1, further corroborating the proposed cascade (Fig. 4g, h). In addition, pharmacological inhibition of AKT1 enhanced BECN1 stability in both assays, leaving about twice the amount of BECN1 at 90 min compared to control (Fig. 4i–l). In order to verify that reduced SKP2 activity drives autophagy in our cellular set-up, we used siRNA and determined the degradation of long-

lived proteins[43]. Targeting SKP2 increased proteolysis, but not in the presence of the autophagy inhibitor 3-methyladenine (3-MA, Fig. 4m). Furthermore, we observed decreased levels of P62 (Supplementary Fig. 1g), which typically goes along with enhanced autophagy[43]. All these data are consistent with an inhibitory effect of SKP2 on autophagy regulated by a phosphorylation cascade involving FKBP51, PHLPP and AKT1.

**Small molecule inhibitors of SKP2 increase BECN1 stability.** Next, we compared known small molecule inhibitors of SKP2[44–46] and found increased stability of BECN1 after a 90 min exposure to SMER3 (leaving about twice as much BECN1 at 60 and at 90 min compared to vehicle control, $p < 0.001$), and SMIP004 in particular (virtually no degradation vs. 78% decrease at 90 min, $p < 0.001$),

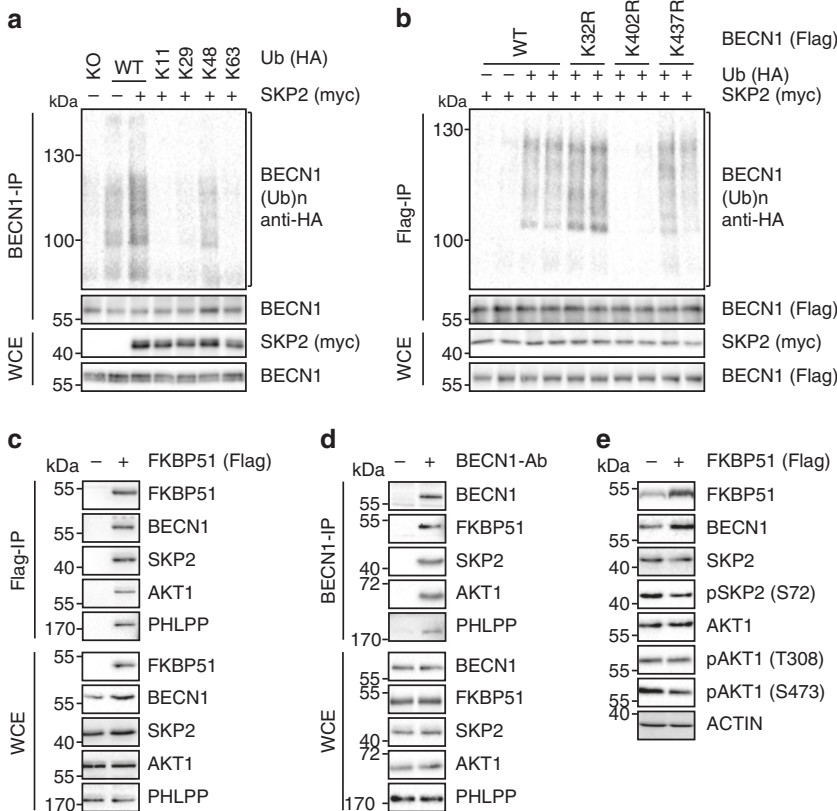

**Fig. 3 SKP2 executes K48-linked poly-ubiquitination at K402 of BECN1. a**, **b** SKP2 attaches K48-linked poly-ubiquitin at K402 of BECN1. HEK293 cells were transfected with plasmids expressing either WT HA-tagged ubiquitin or mutant forms featuring only one lysine together with an SKP2 expressing plasmid or vector control (**a**). In **b**, cells were transfected with plasmids expressing BECN1, either WT or the indicated mutants, myc-SKP2 and HA-ubiquitin. Ubiquitination of BECN1 was determined after immunoprecipitation. **c–e** Association of FKBP51 and BECN1 with regulatory kinases. HEK293 cells were transfected with FKBP51-Flag expressing plasmid (**c**, **e**) and protein extracts were used for western blot analysis after 72 h of cultivation (**e**) or immunoprecipitation of FKBP51 (**c**). In **d**, endogenous BECN1 was precipitated from cell extracts. Blot collections are provided as a Source Data file.

while compound C1 had no effect (Fig. 5a, Supplementary Fig. 2a). The pulse-chase assay confirmed SMIP004 as the most efficient compound in stabilising BECN1 leaving threefold more BECN1 protein after 90 min (Fig. 5b). Furthermore, SMIP004 reduced ubiquitination of BECN1 most efficiently (Fig. 5c, last lane). The long-lived protein assay revealed that SMER3 and SMIP004 enhanced protein half-life, while C1 showed no effect (Fig. 5d). Starvation by cultivating the cells in HBSS, which we confirmed to induce autophagy (Supplementary Fig. 2b-d), did not elicit a greater effect than SMIP004 in the long-lived protein assay (Fig. 5d). Furthermore, SMIP004 most profoundly affected not only BECN1, but also LC3B-II/I and P62 (example blot in Fig. 5e, quantifications in Supplementary Fig. 2e-g) that are established markers of autophagy. To analyse autophagy more thoroughly, it is mandatory to include assays that address autophagic protein turn-over, commonly referred to as autophagic flux[43]. Thus, we subjected SMIP004 to a flux assay with the specific inhibitor of vacuolar H + -ATPases bafilomycin A1 (BafA1), which interferes with lysosome acidification and thus degradation of autophagosome cargo[43]. We revealed a stimulatory effect of SMIP004 on the autophagic flux (Fig. 5f, Supplementary Fig. 2b, c). In addition, it has been shown that ATG14 oligomerizes to dimers and tetramers, which is essential for autophagy by promoting STX17 binding and autophagosome-lysosome fusion[6]. Using capillary-based electrophoresis allowing for better resolution at high molecular weight we observed ATG14 oligomerization to 7–8mers, which was enhanced by SMIP004 (Supplementary Fig. 2h, i). Thus, SMIP004 was used as SKP2-inhibitor (SKP2i) in all subsequent experiments.

**MERS-CoV decreases autophagy**. Results up to this point suggested that activity of BECN1 and thereby autophagy is limited by SKP2, which in turn is activated by AKT1 ([41] and Figs. 3, 4). Since MERS-CoV has been shown to enhance phosphorylation of AKT1[22], it might reduce autophagy. Infection of VeroB4 cells with MERS-CoV facilitated increased phosphorylation of SKP2 at S72 (Fig. 6a, Supplementary Fig. 3a), with a concomitant decrease of cellular levels of BECN1 (Fig. 6a, Supplementary Fig. 3b) and enhanced K48-polyubiquitinylation of BECN1 (Fig. 6b). Infection with MERS-CoV further led to an increase of the total number of phagocytic vesicles (sum of AL + AP) per cell (Fig. 6c, d). However, the number of successfully formed AL was reduced significantly indicating that AP can form but not fuse with lysosomes when cells are infected. A fusion block was also evidenced by the significantly reduced ATG14 oligomerization, essential for AP-lysosome fusion[6], in infected cells (Fig. 6e, f) and by the increase of the autophagy target P62 (Supplementary Fig. 3c). The block of vesicle fusion was further tested by the use of BafA1: as expected, LC3B-II/I levels were not further enhanced upon treatment with BafA1 (Fig. 6g). The fusion between AP and lysosome membranes is facilitated by the interaction between tSNARE proteins STX17 (AP-bound) and VAMP8 (lysosome-bound). This interaction is mediated by SNAP29[47] (Supplementary Fig. 3d). Co-immunoprecipitation revealed that MERS-CoV infection leads to a reduced interaction of STX17 with SNAP29 and VAMP8 (Fig. 6h). This further supports the conclusion that MERS-CoV infection causes a block of AP-lysosome fusion.

Since the results so far raised the possibility that MERS-CoV may benefit from reducing autophagy, we generated VeroB4 cells

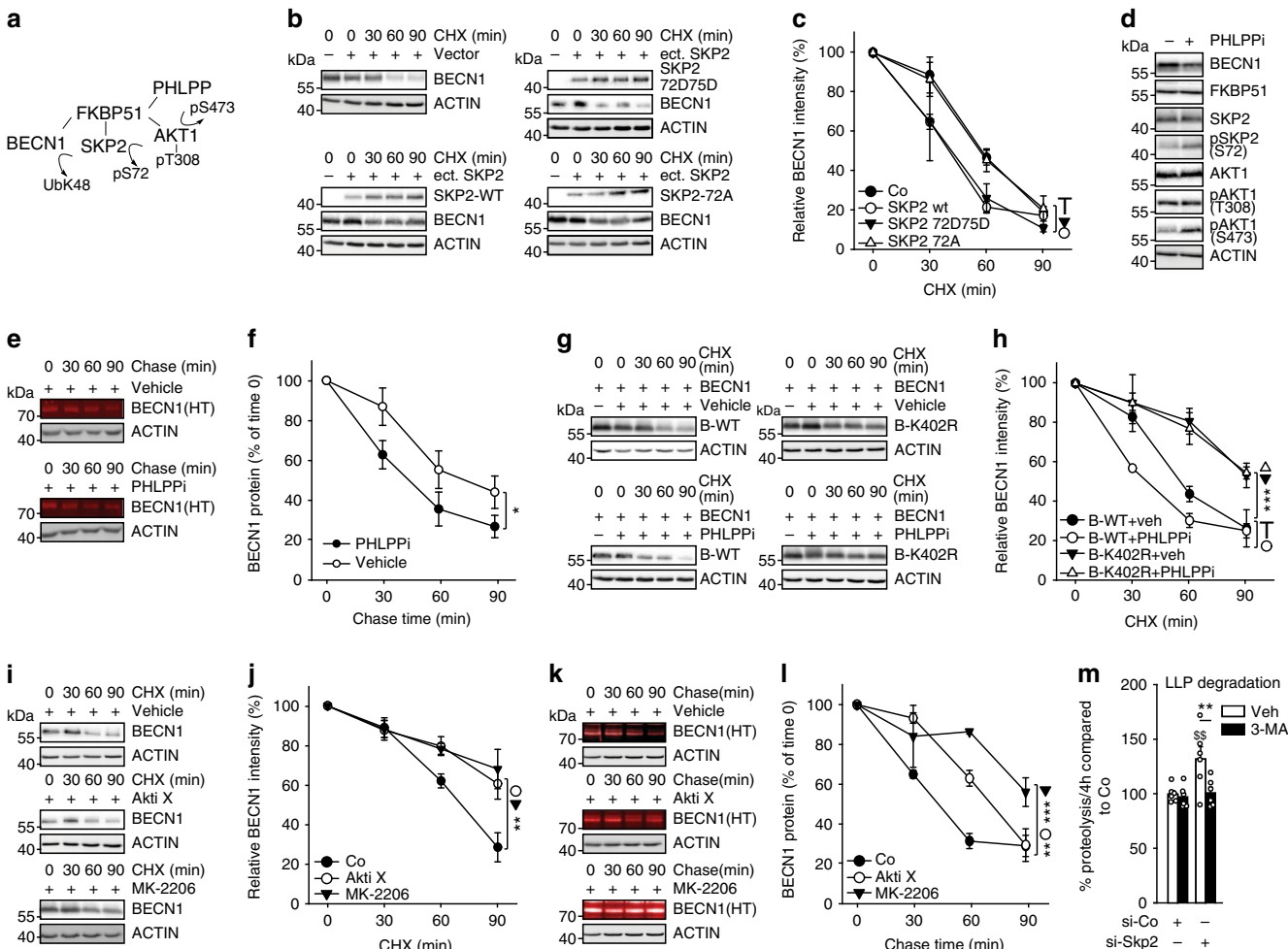

**Fig. 4 SKP2 is controlled by AKT-PHLPP mediated phosphorylation involving FKBP51. a** Model of FKBP51`s kinase associations that regulate SKP2 phosphorylation and activity. The recruitment of PHLPP by FKBP51 leads to lower phosphorylation (S473) and activity of AKT1, which causes decreased phosphorylation and activity of SKP2, thereby stabilizing BECN1. **b, c** BECN1 stability was assessed by the cycloheximide assay (as in Fig. 1c, d) comparing wt SKP2 with the phosphor-null mutant S72A and the phosphomimetic mutant S72D/S75D. **d–f** PHLPP inhibition destabilizes BECN1. HEK293 cells were treated with PHLPPi (NSC117079, 50 μM) for 1 h and cell extracts were probed for the indicated proteins by western blot analysis (**d**). The pulse-chase assay as in Fig. 1e, f was used in **e, f. g–j** The cycloheximide protein stability assay was performed (as in Fig. 1c, d) to evaluate the effect of PHLPPi on K402-BECN1 in comparison to wt BECN1 (**g, h**) and to test the effects of AKT1 inhibitors (AktiX and MK2206) on wt BECN1 (**i, j**). **k, l** The effect of the AKT1 inhibitors on BECN1 was also tested in the pulse chase assay. **m** Reduction of SKP2 by siRNA enhances the half-life of long-lived proteins. HEK293 cells were metabolically labelled with [14C]-valine overnight, chased with fresh media with excess valine overnight and then incubated for 4 h with complete media. The autophagy inhibitor 3-MA (10 mM) was used as control. In all panels, error bars denote the standard error of the mean, derived from $n = 3$ biologically independent experiments. WCE = whole cell extract. *$p < 0.05$, **,$$p < 0.01$, ***$p < 0.001$ (two-way ANOVAs, details in Supplementary Table 1. In **m**, **labels the effect of 3-MA in the si-Skp2 RNA condition, $$labels the effect of the si-Skp2 RNA in the vehicle control. Source data and blot collections are provided as a Source Data file.

with a knockout of the pivotal autophagy gene *ATG5*. Compared to WT cells, infection of VeroB4 *Atg5-KO* cells led to the formation of 52-fold more infectious viral particles (Fig. 7a) while genomic viral RNA copies only increased by 6-fold (Fig. 7b). The efficient formation of DMVs is required for CoV replication and might exploit autophagy or its components[25]. CoV-induced DMV formation is known to depend on viral nonstructural proteins (NSP) 4 and 6[18,48,49]. Ectopic expression of MERS-CoV NSP4 and 6 indeed led to an accumulation of LC3B-II/I and of P62 in the case of NSP6, while NSP4 only had a very minor effect on LC3B-II/I (Fig. 7c). This suggested a block of the autophagic flux by NSP6, which was confirmed by using BafA1 (Fig. 7d), altogether suggesting the MERS-CoV-induced inhibition of autophagic flux to be mediated mainly by NSP6.

CoV have group-specific accessory proteins that can modulate viral replication and immune evasion[50,51]. To assess potential

autophagy-related functions, we overexpressed all MERS-CoV accessory genes 3, 4a, 4b, and 5. Protein 4b and 5, but not 3 and 4a, caused accumulation of P62 and enhanced the level of LC3B-II/I similar to Baf1A, which produced an additional effect in the presence of p3 or p4a, but not p4b or p5 (Fig. 7e, f). Because p4b and p5 are accessory proteins that can be deleted without lethal effects on virus replication, we constructed recombinant MERS-CoVs lacking the respective orfs (Supplementary Fig. 4a, b) to evaluate their effect on autophagy in the context of full virus replication. In comparison to the recombinant MERS-CoV the p4b- and p5-deletion variants showed a decreased accumulation of P62 and LC3B-II/I, suggesting that both proteins contribute to the inhibition of the autophagic flux (Supplementary Fig. 4c, d). The p4b- and p5-deleted viruses as well as the WT control virus grew to higher levels in *Atg5* knockout Vero cells compared to WT cells (Supplementary Fig. 4e, f). However, the p4b and p5-deleted

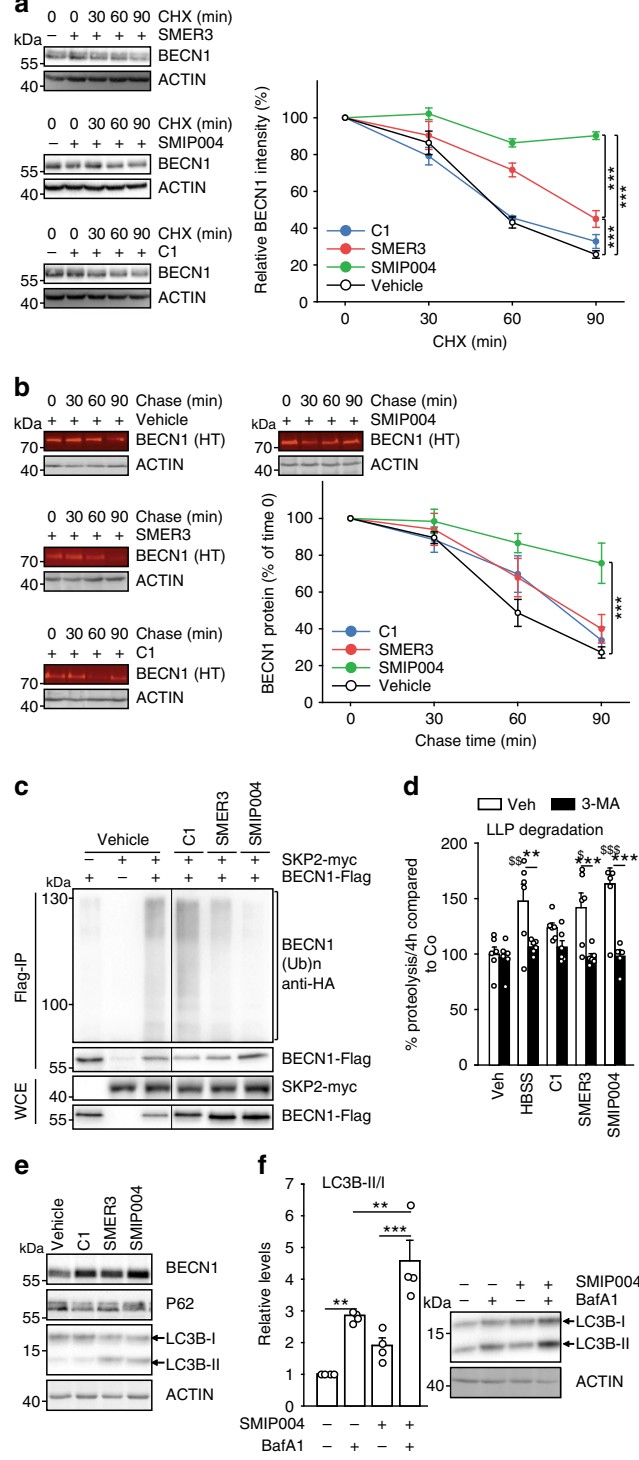

**Fig. 5 SKP2 inhibitors enhance BECN1 protein stability and autophagy. a, b** Evaluation of small compounds inhibiting SKP2. HEK293 cells were exposed to known SKP2 inhibitors for 24 h, followed by additional exposure to cycloheximide (CHX, 30 μg mL$^{-1}$) for the times as indicated or vehicle (**a**). **b** HEK293 cells were transfected with Halo-tagged BECN1 expressing plasmid and exposed to known SKP2 inhibitors and halogenated dye (100 nM, R110Direct) overnight. Cells were transferred to medium without dye and harvested after the indicated times. Labelled BECN1 was determined and quantified. **c** HEK293 cells were transfected with control vector (−) or plasmids expressing BECN1-Flag, HA-ubiquitin and myc-SKP2. Cells were exposed to known SKP2 inhibitors (C1, 3 μM; SMER3, 5 μM; SMIP004, 10 μM) for 24 h, BECN1 was precipitated from cell extracts and its ubiquitination and levels were determined by western blotting. **d** SKP2 inhibitors impact long-lived proteins. HEK293 cells were metabolically labelled with [$^{14}$C]-valine overnight, exposed to the indicated drugs or starvation (HBSS = Hank's balanced salt solution) and the degradation of proteins was determined 4 h after withdrawal of [$^{14}$C]-valine completed media. The autophagy inhibitor 3-MA (10 mM) was used as control. **e, f** Assessment of autophagy markers upon exposure to SKP2 inhibitors. VeroB4 cells were exposed to the indicated drugs or vehicle and the levels of P62 and LC3B-II/I were determined by western blotting (**e**). The most efficient drug, SMIP004, was analyzed in the flux assay with bafilomycin A1 (BafA1; **f**). In all panels, error bars denote the standard error of the mean, derived from n = 3 biologically independent experiments. WCE = whole cell extract. *,$p < 0.05$, **,$$p < 0.01$, ***,$$$p < 0.001$ (two-way ANOVAs, details in Supplementary Table 1). In **d**, *labels refer to the effects of 3-MA, $labels to the drug effects in comparison to vehicle. Source data and blot collections are provided as a Source Data file.

significant reduction of viral replication (by about 250-fold, Fig. 8a, Supplementary Fig. 5a). To explore the relevance of SKP2 inhibition on viral infection in more general terms, we also tested SKP2i in Sindbis virus (SINV) replication. It is known that SINV induces autophagy but that its replication levels are unaffected by it[52]. We observed that treatment with SKP2i caused a moderate decrease of SINV replication (Supplementary Fig. 5b, c). SKP2i may thus be exploitable as a broader therapeutic antiviral principle.

As high concentrations of SKP2i inhibit the cell cycle[44], it was mandatory to control for potential cell cycle effects in VeroB4 cells that may partly account for the observed inhibition of MERS-CoV replication. FACS-based analysis of VeroB4 cells revealed effects on cell proliferation occur only at concentrations of SKP2i beyond 10 μM (Supplementary Fig. 6). Accordingly, while SKP2i enhanced the levels of BECN1 already at 1 μM, no significant increase in the cell cycle promoter P27 was observable up to 100 μM (Supplementary Fig. 7a).

Our data strongly suggest that of the potential mechanisms of SKP2i, it is the increase of BECN1-directed autophagy that reduces MERS-CoV multiplication. If this is correct, other ways of enhancing BECN1's effect on autophagy should elicit similar effects. Therefore, we used two recently introduced tools to enhance BECN1 function in autophagy: the BH3 mimetic ABT-737 and a BECN1-derived peptide[53,54]. These compounds, in contrast to SKP2i, do not change the levels of BECN1, but increase autophagy by redirecting BECN1 function towards the autophagy pathway, which we confirmed in VeroB4 cells (Supplementary Fig. 7b–d, g). Enhanced autophagy was indicated by the increase of LC3B-II/I (Supplementary Fig. 7e, g) and the reduction of P62 (Supplementary Fig. 7f, g). This was paralleled by inhibition of MERS-CoV replication by up to 60-fold in the case of the BECN1-peptide (TAT-B) (Figure Supplementary Fig. 7h) and up to 10-fold in the case of ABT-737 at 48h p.i. (Supplementary Fig. 7i). These results are in accordance with the notion that increasing BECN1

viruses showed overall an up to 10-fold decreased replication in both WT and *Atg5* knockout cells compared to WT virus suggesting a p4b- and p5-dependent attenuation of virus replication that is independent of ATG5-directed autophagy.

**SKP2 inhibition reduces MERS-CoV replication.** The influence of MERS-CoV infection on SKP2 phosphorylation, BECN1 degradation and its inhibition of the autophagic flux encouraged us to test if SKP2 inhibitors (such as SKP2i) may limit MERS-CoV amplification in infected cells. Indeed, SKP2i caused

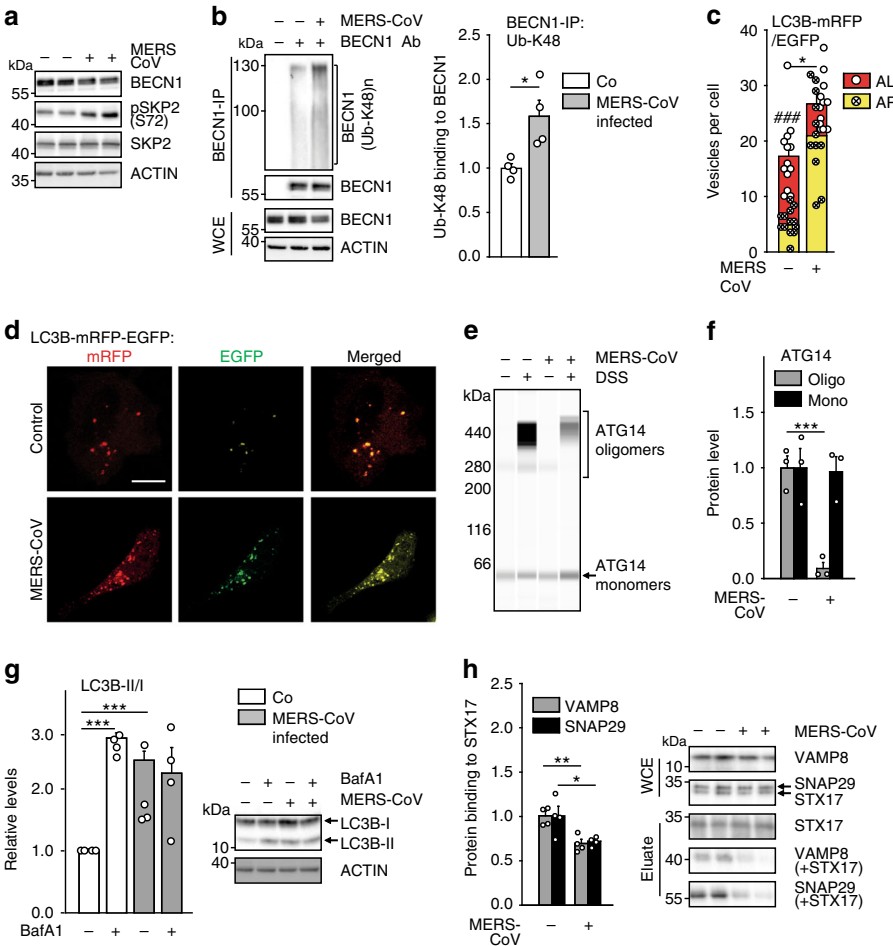

**Fig. 6 MERS-CoV blocks autophagic flux. a** MERS-CoV decreases BECN1 and increases pSKP2. VeroB4 cells were infected with MERS-CoV (MOI = 0.001) or mock-infected and harvested 48 h later. Representative western blots are shown, quantification in Supplementary Fig. 3a, b. **b** MERS-CoV increases K48-linked poly-ubiquitination of BECN1. BECN1 was immune-precipitated from cell extracts 48 h p.i. and its ubiquitination determined by a ubiquitin K48 linkage specific antibody. **c**, **d** MERS-CoV blocks fusion of AP with lysosomes. VeroB4 cells were transfected with tandem fluorescent-tagged LC3B (mRFP and EGFP) and infected with MERS-CoV (MOI = 0.001). Twenty-four hours later, cells were fixed and analyzed for fluorescence. Vesicles with both green and red fluorescence (autophagosomes, AP) and with red fluorescence only (autolysosomes, AL) were counted. **d**, representative images, scale bar 25 μm. **e**, **f** MERS-CoV decreases ATG14 oligomerization. VeroB4 cells were infected with MERS-CoV (MOI = 0.001), cross-linked with disuccinimidyl suberate (DSS, 75 μM) 48 h p.i. for 30 min and harvested. ATG14 homo-oligomerization was examined after western blotting (ProteinSimple). **g** MERS-CoV decreases autophagic flux. VeroB4 cells were infected with MERS-CoV (MOI = 0.001) and incubated with bafilomycin A1 (BafA1, 0.1 μM) for 2 h before samples were taken at 48 h p.i.. The ratios of LC3B-II/I were determined by western blotting. **h** MERS-CoV affects SNARE protein interactions. VeroB4 cells were infected with MERS-CoV (MOI = 0.001), cross-linked with disuccinimidyl suberate (DSS, 75 μM) 48 h p.i. for 30 min and harvested. The SNARE complex protein STX17 was immunoprecipitated and the eluate was probed for interacting VAMP8 and SNAP29 by western blotting (the quantification represents the bands detected at the combined molecular weights of the cross-linked proteins, i.e. STX17 + VAMP8 and STX17 + SNAP29). In all panels, error bars denote the standard error of the mean. derived from n = 3 biologically independent experiments for **b**, **f**, **g**, **h**, and n = 13 (control) or n = 12 (CoV) different cells for **c**. WCE = whole cell extract of vehicle exposed cells, i.e. no cross-linking. *p < 0.05, **p < 0.01, ***p < 0.001 (t-test in **b**, **c**, **f**, **h**, two-way ANOVA in **g**, details in Supplementary Tables 1, 2). Source data and blot collections are provided as a Source Data file.

acts through enhancing autophagy as the relevant mechanism for the antiviral effects of SKP2 in vitro.

To determine whether SKP2i relieves the antagonistic effects of MERS-CoV infection on autophagy, we performed flux assays using either BafA1 or the tandem-labelled LC3B. While BafA1 did not change LC3B lipidation in infected cells as evidenced by constant levels of LC3B-II/I in the absence of SKP2i (as before, Fig. 6g gray bars), it increased LC3B-II/I in the presence of SKP2i (Supplementary Fig. 7j). The fluorescence-based assay revealed that SKP2i enhanced the number of AP while decreasing the number of AL (Fig. 8b, c). Furthermore, SKP2i enhanced ATG14 oligomerization (Fig. 8d, e) and the interaction of autophagic SNARE complex proteins (STX17 with VAMP8 or SNAP29), in MERS-CoV infected cells (Fig. 8f), thereby counteracting the

effect of MERS-CoV (see above, Fig. 6h). Thus, SKP2i inhibits MERS-CoV production, most likely by enhancing BECN1 levels, which might also account for its effect on late steps of autophagy[55]. Since very little interaction between BECN1 and STX17 was detected using immunoprecipitation, this effect of BECN1 may require interaction with ATG14[6].

These results encouraged us to test other potential inhibitors of SKP2. Autophagy-inducing, FDA-approved drugs and drugs from clinical trials were of particular interest as these could become more readily available for treatment in humans. Of note, valinomycin (VAL), which has been shown to target SARS-CoV in vitro[45,56], is also known to act as an SKP2 inhibitor[44].

Of the chosen drugs (Table S3), niclosamide (NIC) and VAL were efficient in enhancing BECN1, LC3B-II/I and autophagic

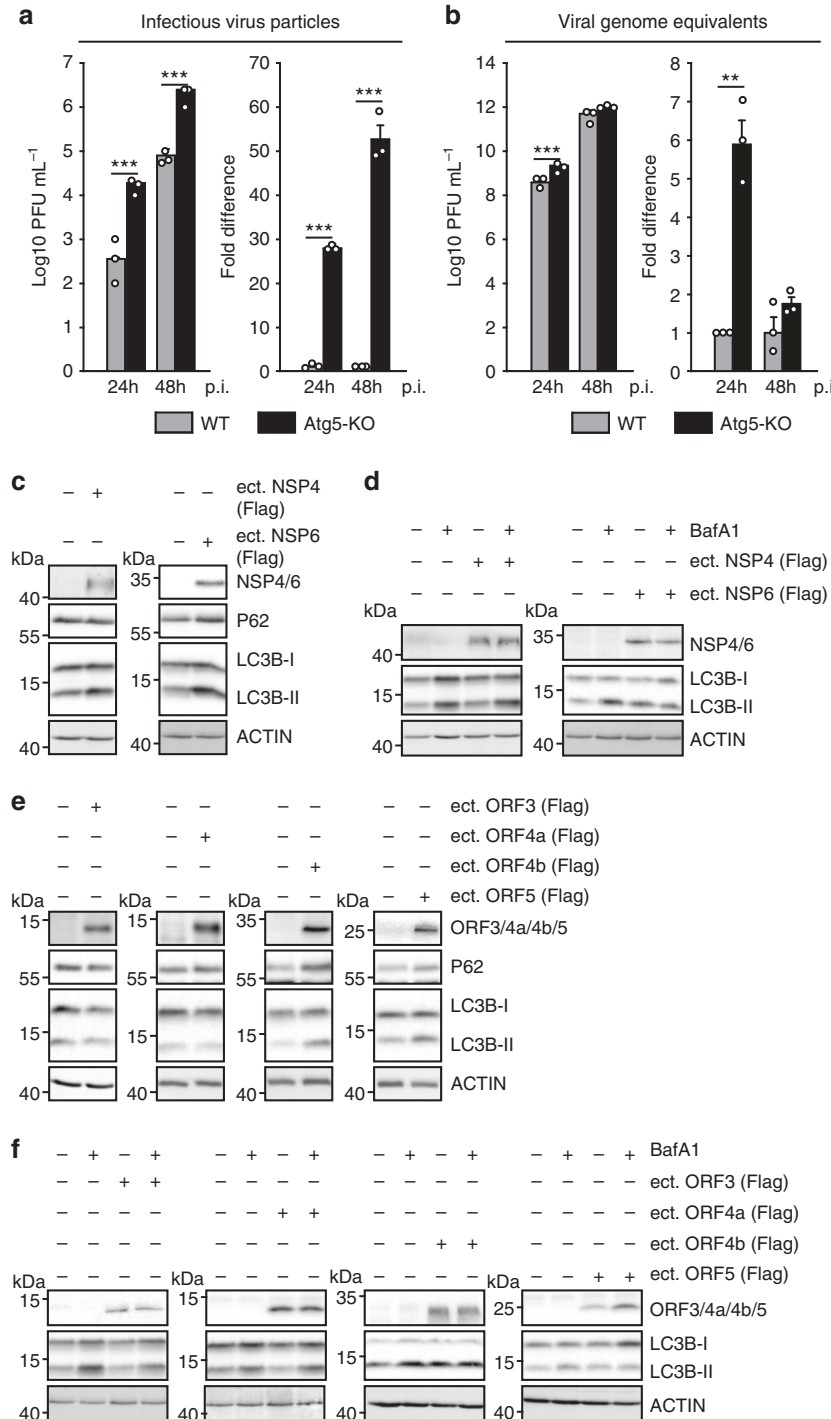

**Fig. 7 Mutual influence of MERS-CoV and autophagy. a**, **b** Deletion of *ATG5* in VeroB4 cells facilitates MERS-CoV replication. VeroB4 wt or *Atg5* knockout cells were infected with MERS-CoV (MOI = 0.001). Plaque forming units (PFU, **a**) and genome equivalents (GE, **b**) per ml were determined by plaque assay or quantitative real time RT-PCR, at 24 and 48 h p.i.. Fold difference and absolute numbers per ml are displayed. In all panels, error bars denote the standard error of the mean, derived from $n = 3$ biologically independent experiments. $**p < 0.01$, $***p < 0.001$ ($t$-tests, details in Supplementary Table 2). **c–f** MERS-CoV NSP4, NSP6 (**c**, **d**) p3, p4a, p4b, and p5 (**e**, **f**, all proteins were Flag-tagged at the N-terminus) were transiently expressed in VeroB4 cells and the indicated proteins were determined by western blotting after 72 h. BafA1 (0.1 μM) was added 2 h before harvesting to assess the autophagic flux (**d**, **f**). Western blots are representative of three independent experiments. Source data and blot collections are provided as a Source Data file.

flux (Fig. 9a, Supplementary Fig. 8a, b, Supplementary Table 3) and comparable to SKP2i (see Fig. 5e, f, Supplementary Fig. 2e, f). NIC and VAL reduced MERS-CoV multiplication by up to 1000-fold at 48 h p.i. (Fig. 9b, Supplementary Fig. 9a–c). Thus, NIC and VAL were characterized further. Both drugs enhanced ATG14 oligomerization about twofold (Supplementary Fig. 9d, e) and

increased the number of autolysosomes more than twofold (Supplementary Fig. 9f, g). NIC also affected the autophagic flux in the infected cells similar to SKP2i (Supplementary Fig. 9h, i). Therefore, these three compounds were tested for their dose-effect relationship in a MERS-CoV infection experiment. All compounds showed high efficacy, ($I_{\mathrm{maxSKP2i}} = 28,000$ fold,

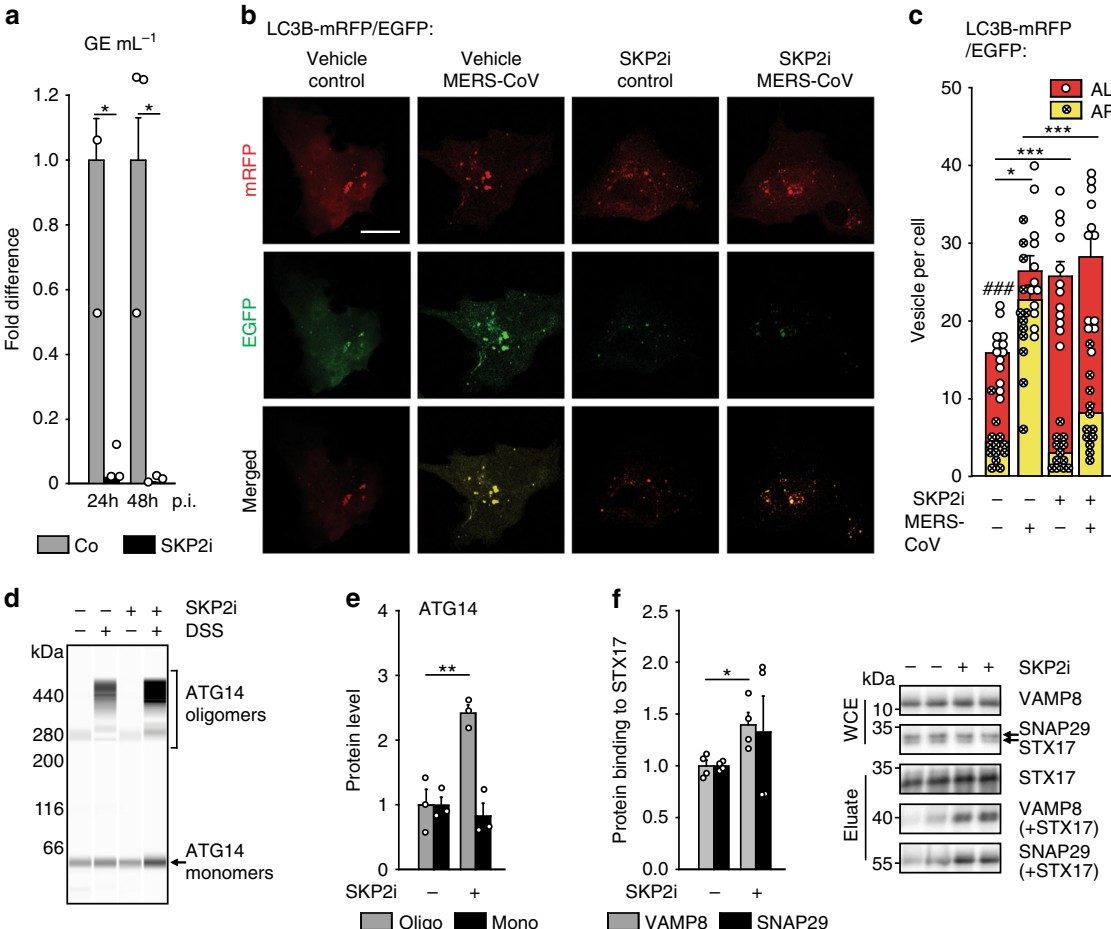

**Fig. 8 SKP2 inhibition reduces replication of MERS-CoV and its effects on autophagy. a** The SKP2 inhibitor (SKP2i) efficiently inhibits MERS-CoV replication. VeroB4 cells were infected with MERS-CoV (MOI = 0.001) and treated with SKP2i (10 μM) or DMSO (Co = control). MERS-CoV GE were determined by real-time RT-PCR at 24 and 48 h p.i., data are presented as fold difference (black bars, SKP2i condition) in comparison to control (gray bars, Co = DMSO condition). Raw data presenting GE ml^-1 is shown in Fig. S5A. **b, c** The SKP2-inhibitor (SKP2i) restores autophagic flux in MERS-CoV-infected cells. VeroB4 cells were transfected with mRFP-EGFP-tagged LC3B, infected with MERS-CoV (MOI = 0.001) and treated with SKP2i (10 μM) or vehicle for 24 h. **b** Representative images (scale bar 25 μm). **c** The numbers of vesicles with both green and red fluorescence (autophagosomes, AP) and with red fluorescence only (autolysosomes, AL) were counted 24 h p.i.. **d–f** SKP2i enhances protein interactions indicative of autolysosome formation in MERS-CoV-infected cells. VeroB4 cells were infected with MERS-CoV (MOI = 0.001), treated with SKP2i for 48 h, cross-linked with disuccinimidyl suberate (DSS, 75 μM) 48 h p.i. for 30 min and harvested. ATG14 homo-oligomerization was assessed after western blotting (**d**) and quantified (**e**). **f** The SNARE complex protein STX17 was immunoprecipitated and the eluate was probed for interacting VAMP8 and SNAP29 by western blotting and quantified as in Fig. 6h. In all panels, error bars denote the standard error of the mean, derived from n = 3 (**a, e**) or n = 4 (**f**) biologically independent experiments, and n = 13 (**c**, vehicle, non-infected) or n = 12 (all other conditions in **c**) different cells. *p < 0.05, **p < 0.01, ***,###p < 0.001 (**c**, *,***refer to the statistical difference between the numbers autolysosomes, ###to the difference between the total numbers of fluorescing vesicles). Two-way ANOVA was performed in **c**, t-tests in **a**, **e**, **f**, details in Supplementary Tables 1, 2. Source data and blot collections are provided as a Source Data file.

$I_{maxNIC} = 23,000$ fold and $I_{maxVAL} = 93,000$ fold; Fig. 9c, Supplementary Fig. 10) and moderate to high potency ($IC_{50SKP2i} = 9$ μM, $IC_{50NIC} = 0.32$ μM and $IC_{50VAL} = 84$ nM).

## Discussion

The present work reveals BECN1 as a target of the E3 ligase SKP2[37,38,46]. SKP2 executes K48-linked ubiquitination at K402 of BECN1, resulting in proteasomal degradation. This action of SKP2 is counteracted by FKBP51 that regulates SKP2 phosphorylation through forming regulatory complexes with AKT1 and PHLPP. Inhibition of SKP2 increases BECN1 levels, enhances the assembly of lysosomal SNARE proteins and autophagy, and efficiently reduces the replication of MERS-CoV (Fig. 9d).

FKBP51 is a pivotal element of molecular feedback loops of the cellular and physiological stress reaction. It increasingly emerges

as a scaffolder that chaperones various regulatory protein interactions[3]. Here, FKBP51 appears to recruit SKP2 in its inactive form to BECN1, probably by virtue of its interaction with the kinase AKT1 and the phosphatase PHLPP. Similarly, FKBP51-directed protein interactions increase BECN1 phosphorylation[2], decrease phosphorylation and activity of AKT1[31]. Thus, FKBP51 impacts BECN1 at least via two mechanisms, phosphorylation of BECN1 itself[2] and of its regulatory E3 ligase SKP2. Both mechanisms involve AKT1, which exhibits yet another way of influencing BECN1 by phosphorylating USP14, which removes K63-linked ubiquitins from BECN1[57].

In addition to K48-linked poly-ubiquitination by SKP2, BECN1 undergoes ubiquitination also by other E3 ligases, at different sites, and with different types of poly-ubiquitination linkage[12,58,59]. NEDD4 produces both K63- and K11-linkages at yet to be determined sites of BECN1[9]. Decoration of BECN1 with

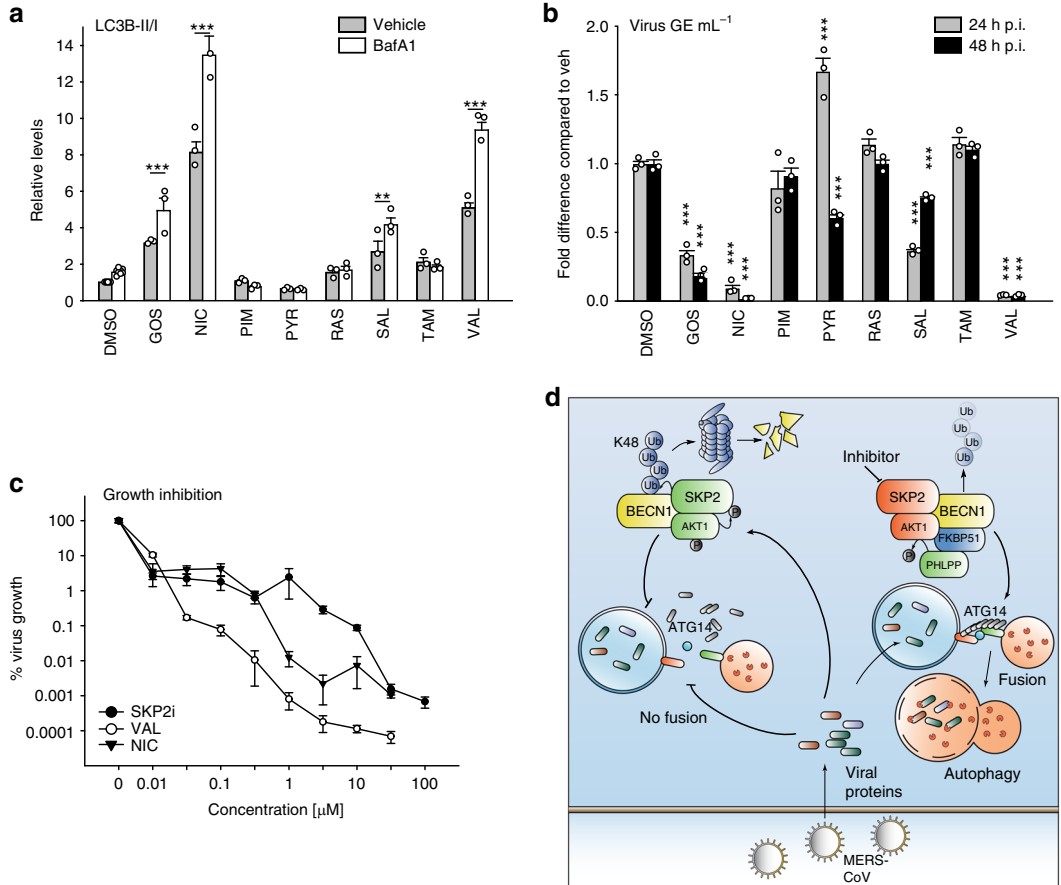

**Fig. 9 Some potential SKP2 inhibitors affect autophagy and viral replication. a** VeroB4 cells were treated with various FDA-approved drugs (for abbreviations and concentrations see Supplementary Table 3) and cotreatment with Bafilomycin A1 (0.1 μM) was performed to evaluate the autophagic flux. The graph provides the means + SEM of three independent experiments, representative western blots are shown in Supplementary Fig. 8b. **b** VeroB4 cells were infected with MERS-CoV (MOI = 0.001), treated with the indicated drugs, and MERS-CoV genome copies were determined by RT-PCR 24 h and 48 h p.i.; data are presented as (log10) fold difference in relation to the DMSO control. Unprocessed data and PFU results are presented in Supplementary Fig. 9a–c. **c** Concentration-dependent inhibition of MERS-CoV replication by the SKP2-inhibitor (SKP2i), valinomycin (VAL) und niclosamide (NIC). VeroB4 cells were infected with MERS-CoV (MOI = 0.001) and treated with drug. MERS-CoV genome copies were determined by real-time RT-PCR at 48 h p.i., data present the percentage of remaining MERS-CoV genomes in comparison to vehicle treatment. In all panels, error bars denote the standard error of the mean, derived from $n = 3$ or $n = 6$ (**c**, no drug condition) biologically independent experiments. **$p < 0.01$, ***$p < 0.001$ (two-way ANOVA in **a**, one-way ANOVA in **b**, details in Supplementary Table 1). **d** Summary scheme: SKP2 leads to K48-linked poly-ubiquitination and thus degradation of BECN1. The effect of SKP2 can be diminished in two ways, either by chemical inhibition or by FKBP51, which scaffolds protein interactions ultimately leading to SKP2 inactivation through inhibiting its phosphorylation. Both scenarios enhance autophagy, which involves ATG14 oligomerization (probably 7–8mers) as recently described essential step in functional autophagy[6]. MERS-CoV reduces autophagy through distinct viral proteins leading to blockade of autophagosome-lysosome fusion and ATG14 oligomerization. Compounds inhibiting SKP2 reinstate autophagy and efficiently reduce viral production. Source data are provided as a Source Data file.

the non-canonical K11-linked ubiquitins by NEDD4 leads to its proteasomal degradation[60]. K63-linked ubiquitination of the BH3 domain of BECN1 at K117 by TRAF6 enhances autophagy, possibly by reducing the interaction of BECN1 with BCL2[35]. AMBRA1 is another positive regulator of autophagy[61]. As a substrate receptor for the DDB1-cullin4 E3 ligase complex[62], it forms K63-linked ubiquitin chains at K437 of BECN1, enhances the association between BECN1 and Vps34/PI3K3C, and thus stimulates the activity of this autophagy promoting lipid kinase[63]. In sum, BECN1 activity is regulated by ubiquitination in multiple ways, both stimulatory and inhibitory.

Very recently, the polyQ domain protein ataxin 3 has been shown to interact with BECN1 and to remove K48-linked poly-ubiquitin, an activity that is competitively inhibited by longer polyQ mutation in a disease protein[64]. Together with the here-presented discovery of SKP2 as K48-linking E3 ligase of BECN1,

this points to the intriguing possibility that inhibiting SKP2 using small molecule inhibitors might counteract deleterious effects of soluble proteins with polyQ expansions that are held responsible for causing neurodegenerative diseases such as Huntington's disease and ataxin 3 in spinocerebellar ataxia[64].

Due to the tremendous variability of viral proteins, there is huge interest in host cell-encoded pathways that could be exploited for fighting viral infection. The present study as well as earlier work suggest autophagy as a pivotal component of virus-host interaction and potential broad-range antiviral target[13,52]. For MERS-CoV replication, it remains to be elucidated whether blocking basal autophagy is mandatory or only auxiliary; nevertheless, the virus encodes at least three proteins that limit autophagy when expressed ectopically, NSP6, p4b and p5. Although the detailed molecular mechanisms are yet to be analysed, MERS-CoV NSP6 may also play a role in inhibiting the expansion of autophagosomes as

shown for other betacoronaviruses[48]. MERS-CoV p4b has phosphodiesterase function inhibiting the activation of RNAse L, which itself can activate autophagy[65]. MERS-CoV p5 is located in the endoplasmic reticulum-Golgi intermediate compartment[51] and was shown to inhibit IFN-beta induction[66], which may constitute another link to autophagy[67].

Deletion of p4b or p5 resulted in reduced MERS-CoV growth. These results differ from previous observations, which may be related to different cell lines and cDNA clone construction strategies[68,69]. In addition, we used a very low MOI and a type I IFN-deficient cell line (VeroB4), which allows to determine minor growth differences. The absence of p4b and p5 during MERS-CoV replication may also result in lower LC3B levels compared to MERS-CoV wildtype-infected but also to the mock-infected control cells. This points to a more complex interaction between virus replication and autophagy; p4b and p5 may counteract a response of the host to viral infection, but the functional relevance of their link to autophagy requires further investigation. Apparently, several CoV proteins collaborate to reprogram the membrane traffic in the cell for DMV formation serving as compartments of viral genome replication and transcription[18,24]. Since autophagy is tightly linked to intracellular membrane traffic and turnover, it appears possible that MERS-CoV blocks autophagy not only to evade degradation but also to improve membrane availability.

It appears that most of the potential treatments arising from other observations of reduction of viral replication upon autophagy stimulation are not easily applicable in vivo because of side effects and inefficient peptide delivery[70]. Examples are rapamycin and vitamin D which restricted the replication of HIV in primary human macrophages[71], and the autophagy-inducing BECN1-derived peptide which limited the replication of a number of viruses in vitro[53]. From the present results, therapeutic induction of autophagy by inhibition of SKP2 emerges as a promising approach. We should note that all the substances tested here have or may have additional targets. While we cannot exclude the possibility that for each of the compounds it is this additional target that is responsible for the antiviral effect, we consider this scenario as less likely; this assessment is also based on the observation of the antiviral effects of compounds targeting BECN1, which we establish here as client of SKP2. Nevertheless, future experiments should assess the effect of these drugs in autophagy-defective cells such as the Atg5 knockout cells.

It further remains to be investigated whether inhibiting SKP2 will affect all autophagy-sensitive viruses. SINV, which was not overtly influenced by *Atg5* knockout despite indications of degradation of viral components through autophagy[72], reacted modestly to SKP2i. The reason for this is unknown; it may relate to the involvement of other immune pathways influencing SINV, for example the JAK/STAT pathway[73] or stress granule formation[74], or to autophagy functioning in the absence of *Atg5*[75]. FDA-approved drugs known to act as SKP2 inhibitors are available[44]. We show that some of these drugs are effective in reducing MERS-CoV replication in vitro, for example niclosamide and pyrvinium pamoate. Even though there are reports of low absorption of these drugs[76], others found considerable absorption and serum levels of niclosamide reached 1-20 μM[77], concentrations we show here to efficiently limit MERS-CoV replication. In addition, inhalable formulations of niclosamide have been developed[78]. SKP2-targeting compounds also may have potential against other infections and other clinical conditions that are influenced by autophagy induction[79,80].

## Methods

**Chemicals**. The following chemicals were used for treatment of cells: BafA1 (Alfa Aeser, J61835), 17-AAG (Sigma-Aldrich, A8476), 6-AN (Sigma-Aldrich, A68203), AMI (Sigma-Aldrich, 1019701), BRO (Sigma-Aldrich, 1076501), CEL (Sigma-

Aldrich, PZ0008), CLO (Sigma-Aldrich, C7291), ETO (Sigma-Aldrich, E1383), (MET (Sigma-Aldrich, M0605000), PAR (Sigma-Aldrich, P9623), RAP (Tocris, 1292), SMER3 (Tocris, 4375), ABT-837 (Selleckchem, S10002), Tat-B (Anaspec, AS65467), Tat-scr. (Anaspec, AS65468), GOS (Sigma-Aldrich, G8761), HAL (Tocris, 0931), MIN (Sigma-Aldrich, M9511), NIC (Sigma-Aldrich, N3510), PIM (Sigma-Aldrich, P1793), PRT (Tocris, 0610), PER (Sigma-Aldrich, SML0120), PEP (Sigma-Aldrich, P6402), PYR (Sigma-Aldrich, 1592001), RAS (Sigma-Aldrich, SML0124), SEC (Sigma-Aldrich, SML0055), SAL (Sigma-Aldrich, S4526), TAM (Sigma-Aldrich, T5648), TOP (Tocris, 4562), VAL (Sigma-Aldrich, V0627), PHLPPi NSC117079 (GlixxLabs, GLXC-03994), $C^{14}$-Valine (Perkin Elmer, NEC291EU050UC), Akti X (Cayman Chemical, 14863) MK-2206 HCl (Cayman Chemical, 11593).

**Co-Immunoprecipitation (CoIP) with crosslinker**. CoIP of endogenous STX17 or BECN1 were performed in VeroB4 cells. The PBS-washed cell pellet was incubated with 75 μM DSS (disuccinimidyl subernate, Thermo Fisher Scientific, 21655) or corresponding vehicle (DMSO) in PBS. Crosslinking was performed for 30 min at room temperature followed by 2 h at 4 °C while rotating. After washing with PBS, crosslinking was quenched in tris-buffered saline (pH 7.0) for 20 min at 4 °C. Cells were lysed in CoIP-buffer containing 20 mM Tris-HCl pH 8.0, 100 mM NaCl, 1 mM EDTA, 0.5% IGEPAL CA-630 complemented with protease inhibitor cocktail (Sigma, P2714). This was followed by incubation on an overhead shaker for 20 min at 4 °C. 1.2 mg of lysate was incubated with 2.5 μg STX17 or BECN1 antibody overnight at 4 °C. 20 μl of BSA-blocked Protein G Dynabeads (Invitrogen, 100–03D) were added to the lysate–antibody mix followed by 3 h incubation at 4 °C. The beads were washed three times with PBS and protein-antibody complexes were eluted by boiling for 5 min at 95 °C in Laemmli loading buffer. Fifteen microgram of the cell lysates or 5 μl of the immunoprecipitates were separated by SDS-PAGE.

**Toxicity assays**. LDH assay. To assess membrane disruption and cell death the release of lactate dehydrogenase (LDH) into the growth medium of HEK293 or VeroB4 cells was measured. The assay (LDH cytotoxicity detection system) was carried out according to the manufacturer's protocol (Clontech, Mountain View, CA, USA). As positive control 0.02% Triton X-100 was added to the medium 1–2 h prior performing the assay; empty wells were used with medium only, as negative control.
MTT assay. HEK293 or VeroB4 cells were incubated in the presence of 0.5 mg mL$^{-1}$ tetrazole 3-(4,5-dimethylthiazol-2-yl-)-2,5-diphenyltetrazolium bromide (MTT) for 4 h at 37 °C and 5% $CO_2$. Read-out of MTT assay was carried out as described previously[81].

**Western blot analysis**. Protein extracts were obtained by lysing cells in RIPA buffer (150 mM NaCl, 1% IGEPAL CA-630, 0.5% Sodium deoxycholate, 0.1% SDS 50 mM Tris (pH8.0)) freshly supplemented with protease inhibitor (Merck Millipore, Darmstadt, Germany), benzonase (Merck Millipore), 5 mM DTT (Sigma Aldrich, Munich, Germany), and phosphatase inhibitor (Roche, Penzberg, Germany) inhibitor cocktail. Proteins were separated by SDS-PAGE and electro-transferred onto nitrocellulose membranes. Blots were placed in Tris-buffered saline, supplemented with 0.05% Tween (Sigma Aldrich) and 5% non-fat milk for 1 h at room temperature and then incubated with primary antibody (diluted in TBS/0.05% Tween) overnight at 4 °C. The following primary antibodies were used: Actin (1:5000, Santa Cruz Biotechnologies, sc-1616), Beclin-1 (1:1000, Cell Signaling Technology, #3738), K48-linkage Specific Polyubiquitin (D9D5) Rabbit mAb (Cell Signaling #8081, 1:500), PI3K3C (1:1000, Cell Signaling Technology, #4263), LC3-B (1:1000, Cell Signaling Technology, #3868), SQSTM1/p62 (1:1000, Cell Signaling Technology, #5114), VAMP8 (1:1000, Cell Signaling Technology, #13060), Atg14 (1:1000, Cell Signaling Technology, #5504), STX17 (1:1000, Sigma Aldrich, SAB001204), SNAP29 (1:1000, Sigma Aldrich, SAB2107406), PHLPP1 (1:1000, Millipore, 07–141). The antibody recognizing pSKP2 (S72) was a kind gift from Cell Signaling Technology®.
Subsequently, blots were washed and probed with the respective horseradish peroxidase- or fluorophore-conjugated secondary antibody for 1 h at room temperature. The immuno-reactive bands were visualized either using ECL detection reagent (Millipore, Billerica, MA, USA) or directly by excitation of the respective fluorophore. Determination of the band intensities were performed with BioRad, ChemiDoc MP.
In the case of ATG14 oligomerization, lysates were analyzed by capillary electrophoresis on Wes™ (ProteinSimple) using the 60–440 kDa cartridges.
In general, protein quantification was performed by normalization to the intensity of Actin, which was determined on the same membrane. For quantification of lipidated LC3B, the intensity of LC3B-II was always referred to the signal intensity of the corresponding LC3B-I, following the guidelines for the determination of autophagy[43].

**RIPA lysis of MERS-CoV infected VeroB4 cells**. Whole cell lysates were prepared with RIPA lysis buffer containing 150 mM NaCl, 1% IGEPAL CA-630, 0.5% sodium deoxycholate, 0.1% SDS and 50 mM Tris (pH 8.0). 1% proteinase inhibitor cocktail III [Calbiochem], 0.1% benzonase [Novagen], 0.5% dithiothreitol [0.1 mM] and 1% phosphataseinhibitor [PhosphoSTOP, Roche] were freshly added. For cell lysis, supernatant was discarded and cells were carefully rinsed 3x with pre-cooled 1x PBS. 500 μl of pre-cooled 1x PBS was added and cells were scraped off and

transferred to a microcentrifuge tube. Cells were pelleted by centrifugation at 4 °C and $300 \times g$ for 5 min The supernatant was removed carefully and cells were resuspended in 100 µl RIPA lysis buffer and incubated for 20 min at 4 °C for efficient lysis. Then, SDS loading buffer (4x NuPAGE LDS, ThermoFisher Scientific) was added and samples were boiled at 95 °C for 10 min Samples were stored at −20 °C and processed as indicated in the respective sections.

**Cells**. VeroB4 (DSMZ [German Collection of Microorganisms and Cell Cultures] number-ACC33) were cultivated in Dulbecco's Modified Eagles Medium (DMEM) supplemented with 10% fetal bovine serum (FBS), 1% penicillin/streptomycin, 1% non-essential amino acids, 1% L-glutamine and 1% sodium pyruvate at 37 °C and 5% $CO_2$. HEK293 cells (ATCC, Manassas, VA, USA CRL-1573) were cultivated in DMEM supplemented with 10% FBS, 1% penicillin/streptomycin and 1% sodium pyruvate at 37 °C and 5% $CO_2$. For transfection HEK293 and VeroB4 were detached and resuspended in 100 µL (50 mM HEPES pH 7.3, 90 mM NaCl, 5 mM KCl, 0.15 mM $CaCl_2$) transfected using the Nucleofector II system (program # T020).

**Generation of Atg5 KO VeroB4**. Vero B4 were plated in 6-well plates at a cell density of $5 \times 10^5$/well in VeroB4 medium and transfected with lipofectamine 3000 according to the manufacturer's protocol. For the Atg5 knock-out, the pSpCas9 BB-2A-Puro (PX459) V2.0 plasmid containing Atg5-targeting gRNA (sequence: 5′-TATCCCCTTTAGAATATATC-3′) purchased from GenScript was used. Forty-eight after transfection, 10 µg mL$^{-1}$ puromycin (InVivoGen, ant-pr-1) was added to the medium. After 48 h, the medium was changed to puromycin-free VeroB4 medium and, another 48 h later, cells were sorted into a 96-well plate at a density of one cell per well using the BD FACSMelody cell sorter (BD Biosciences). Cells were maintained in conditioned medium for 1 week until a colony was formed from a single cell. All the colonies were then trypsinized and expanded individually. Atg5 knockout was examined for each clone by western blot analysis using anti-Atg5 purchased from Santa Cruz (sc-133158).

**Virus infection**. For virus infection, cells were seeded in a concentration of $3.5 \times 10^5$ cells mL−1. After 24 h, virus stocks were diluted in serum-free medium according to the desired MOI. For virus adsorption, 1 ml (six-well) virus master mix was added to the cells and incubated for 1 h at 37 °C. When comparing to non-infected cells, heat-inactivated (95 °C, 10 min) virus was used (mock control). After 1 h, the virus dilutions were removed and the wells were washed twice with 1x PBS and refilled with DMEM (supplemented as described above). Samples were taken at the indicated time points. All virus infection experiments were conducted under biosafety level 3 conditions with enhanced respiratory personal protection equipment.

For all SINV infection experiments strain LEIV-Ast03 was used. SINV infections were done using VeroB4 cells applying an MOI = 0.0001 for 16 and 24 h. The SKP2-inhibitor SMIP004 (SKP2i) was applied at a concentration of 10 µM.

**Real-time RT-PCR assay**. Genome equivalents were determined by real-time RT-PCR. Therefore, 75 µl of cell supernatant was lysed in RAV1 lysis buffer (Macherey & Nagel) and heated to 70 °C for 10 min for virus inactivation. Then, viral RNA was extracted according to the manufacturer's instructions. Real-time RT-PCR assays were performed as before[82]. More specifically, the upE assay was applied for the detection of MERS-CoV genome equivalents, the ORF1b assay was used for the detection of MERS-del4b and MERS-del5 genome equivalents. Primers were upE-Fwd (5′-GCAACGCGCGATTCAGTT-3′), upE-Rev (5′-GCCTCTACACGGGAC CCA TA-3′) and 200 nM of probe upE-Prb (6-carboxyfluorescein [FAM])-CTCTTT CACATAA TCGGCCCGAGCTCG-6-carboxy-N,N,N,N′-tetramethylrhodamine [TAMRA]) as well as ORF1b-Fwd (5′-TTCGATGTTGAGGGTGCTCAT-3′), ORF1b-Rev (TCACACCAGTTGA AAATCCTAATTG) and probe ORF1b-Prb (6-carboxyfluorescein [FAM])-CCCGTAAT G CATGTGGCACCAATGT-6-car boxy-N,N,N,N′-tetramethylrhodamine [TAMRA]). For quantification of BECN1-mRNA, primers (forward 5′-CAAGATCCTGGA-3′, reverse 5′-CCGTGTCA-3′) were used as described in[83].

**Plaque assay**. SINV infectious particles were quantified by plaque titration on VeroB4 cells as previously described[84] with minor modifications. Briefly, VeroB4 monolayers were seeded in 24-well plates, incubated with SINV-containing cell culture supernatants and overlaid with 1.2% Avicel in MEM containing 2% FCS and 1% penicillin-streptomycin. After 16 h, 24 h or 72 h, plaques were fixated and visualized by staining with crystal violet and counted.

**Plasmids and construction of MERS-CoV cDNA clones**. For heterologous expression, nsp4 and nsp6 were PCR amplified from MERS-CoV cDNA and cloned into the eukaryotic expression plasmid pCAGGS along with a C-terminal FLAG tag (MERS-nsp4-F: 5′-AGAGAATTCGCCACCATGGCTCCTACATGGTTTAATG C-3′, MERS-nsp4-R: 5′-TCTGCGGCCGCCTATCACTTGTCATCGTCGTCCTTGT AGTCGCC GGCTTGCAACACGCCAGAGGTTATGC-3′, MERS-nsp6-F: 5′-AGAG AATTCGCCACC ATGAGTGGTGTGAGAAAAGTTACATATGG-3′, MERS-nsp6-R: 5′-TCTGCGGCCGCC TATCACTTGTCATCGTCGTCCTTGTAGTCGGCGGC CTGCATAGCAGCAACCTTTAA CAAGG-3′). Construction of the plasmids

expressing the accessory open reading frames (ORFs) 3, 4a, 4b, and 5 was previously described[51].

Construction of a MERS-CoV cDNA clone lacking accessory ORFs: The accessory ORFs (ORF4b or ORF5) were deleted using a two-step markerless Red Recombination system as extensively described[85,86]. Briefly, in a first step a kanamycin resistance gene was amplified by an overhang extension PCR from the pEP-KanS vector (kindly provided by K. Tischer and K. Osterrieder) with primers containing homologous MERS-CoV recombination sites, the desired mutation and an I-SceI restriction site (MERS-delta4b-ABC-Kana-F: 5′-ATT CGCGCAAAGCG AGGAAGAGGAGCCATTCTCAACTAATGATGTTGTCTCCATACGG TCTTT AGGGATAACAGGGTAATCGATTT-3′ and MERS-delta4b-BCD-Kana-R: 5′-TT GTTTATTACCCTGATTGGAAGACCGTATGGAGCAACATCATTAGTTGG AGAATG GCTCCTGCCAGTGTTACAACCAATTAACC-3′; MERS-delORF5_ Kana-F: 5′-TTCTTA TCCCATTTTACATCATCCAGGATTTTAACGAACTGCA GCTCTGCGCTACTATGGTA GGGATAACAGGGTAATCGATTT-3′ and MERS-delORF5_Kana-R: 5′-GATTAGCCTC TACACGGGACCCATAGTAGCG CAGAGCTGCAGTTCGTTAAAATCCTGGATGCCAG TGTTACAACCAATTA ACC-3′). We would like to point out that this design preserves the regulatory sites of ORF5, in contrast to a previous approach[69]. Our rationale was to preserve an equal number of subgenomic mRNAs. *E. coli* GS1783 cells (kindly provided by K. Tischer and K. Osterrieder) containing the MERS-CoV cDNA clone were transfected with the PCR product in order to induce the recombination of the transfer construct into the MERS-CoV backbone. In a second step, the kanamycin selection marker was excised by an inducible in vivo I-SceI cleavage and the second Red recombination event. Rescue and stock production of the recombinant viruses were performed as described elsewhere[85].

The following expression constructs were used for experiments performed in cultured cells: pRK5-FKBP51-Flag, pRK5-FKBP52-Flag[2,87], pRK5-HA-Ubiquitin-WT (Addgene.org, #17608), pRK5-HA-Ubiquitin-KO (Addgene.org, #17603), pRK5-HA-Ubiquitin-K11 (Addgene.org, #22901), pRK5-HA-Ubiquitin-K29R (Addgene.org, #17602), pRK5-HA-Ubiquitin-K48 (Addgene.org, #17605), pRK5-HA-Ubiquitin-K63 (Addgene.org, #17606), Beclin1-HT (Promega, FHC08488), FLAG-HA-USP18 (Addgene.org, #22572), FLAG-HA-USP36 (Addgene.org, #22579), pcDNA3-myc-SKP2 (Addgene.org, #19947), FLAG-TRAF6-wt (Addgene.org, #21624), pcDNA4-Beclin1-HA (FL) (Addgene.org, #24388). The phosphomimetic and phospho-null mutants S72D/S75D and S72A of SKP2 were kind gifts from the authors of[41]. Transfection of the constructs FLAG-HA-USP18 and and FLAG-HA-USP36 each led to the appearance of another band at higher molecular weight in western blots. Irrespective of the exact nature of these bands, specific co-precipitation with FKBP51 was only observable for the bands at the predicted molecular weight (Fig. 1a); these were absent in coprecipitations of BECN1 (Fig. 1b). Figure 1a, b display the bands at the predicted molecular weight.

**Skp2 knockdown**. HEK293 cells were electroporated as described above with skp2-siRNA (sense 5′-GUCGGUGCUAUGAUAAUU-3′; anti-sense 5′-AUUAUAUC AUCGCACCGACTT-3′) or control siRNA (sense 5′-UUCUCCGAACGUGUCA CGUTT-3′; anti-sense 5′-ACGUGACACGUUCGGAGAATT-3′) and harvested after 48 h. Primers were purchased from Biomol (Hamburg, Germany).

**Autophagic flux**. To determine the effect of drug treatment or MERS-CoV infection on autophagic flux, either BafA1 (Alfa Aeser, J61835), a specific inhibitor of vacuolar H+-ATPases interfering with lysosome acidification and thereby degradation of autophagosome cargo, was used or transfection with a plasmid (ptfLC3, Addgene.org, #21074) that expresses LC3 tagged with both GFP (inactivated in autolysosomes) and mRFP (resists inactivation in autolysosomes)[88], following the recently updated guidelines for monitoring autophagy[43]. Transfection in Vero4B cells was performed using Fugene transfection reagent (Promega), followed by MERS-CoV infection (MOI = 1) and drug treatment the next day. Cells were fixed (1% PFA for 1 h) 1 day later and analyzed by laser scanning microscopy (Leica Confocal Sp8). Vesicles were counted by an experimenter blind to the conditions.

**Long-Lived protein degradation assay**. Experiments were performed as described by Klionsky et al.[43]. Cells were incubated with valine[14C] overnight, chased in fresh media with excess valine overnight, and then incubated 4 h with complete medium or starvation media (HBSS/EBSS + 10 mM HEPES, Starv). To check that protein degradation measured was due to autophagic processes, experiments were also performed in the presence of 3-MA (10 mM, which is typically used for the inhibition of autophagy[89]). LLP degradation with SKP2-KD was monitored 48 h after transfection.

**Inhibition of translation by cycloheximide (CHX)**. HEK293 cells were incubated with 20 µg mL$^{-1}$ CHX. MG132 (10 µM), 3.5 µg expression construct for FKBP51-Flag, empty vector pRK5 SV40 MCS[2], TRAF6-Flag (Addgene.org, #21624), SKP2-MYC (Addgene.org, #19947) was used for stimulation or transfection of HEK293 cells.

**Pulse-Chase-Assay using Halo®-tagged BECN1**. HEK293 cell were transfected with 5 µg BECN1-HT expression construct. The pulse-chase assay was performed according to manufacturer's manual using R110Direct as halogenated fluorophore

(100 nM, R110Direct, #G3221, Promega) over-night (pulse), followed by switching the cells to medium without dye (chase).

**Determination of $IC_{50}$.** To estimate the maximal viral inhibition ($I_{max}$) and the drug concentration achieving viral inhibition halfway between baseline and maximum ($IC_{50}$), we first normalized the dose-response curves for each drug by dividing by cell viability, and we then fitted the normalized viral inhibition curves to a sigmoid curve using the Sigmoid and Nonlinear Least Squares functions in R version 3.1.0 (http://www.R-project.org/) as previously described (http://kyrcha.info/2012/07/08/tutorials-fitting-a-sigmoid-function-in-r/). Based on the sigmoid-fitted curves, we then determined numerically the $I_{max}$ and $IC_{50}$.

**Flow cytometry**. Cells were washed in PBS and fixed with cold ethanol (70%) for 30 min at 4 °C. After washing with PBS cells were treated with ribonuclease A (100 μg mL$^{-1}$; Sigma Aldrich) and stained with Vybrant DyeCycle Orange (Invitrogen, V35005) as described by the manufacturer's manual. Cells were analyzed with the FACSCalibur (BD Biosciences). Acquired data were analyzed with FlowJo software (Tree Star).

**Statistical analysis**. When two groups were compared, the student's *t*-test was applied. For three or more group comparisons, one or two-way analysis of variance (ANOVA) was performed, as appropriate, followed by Tukey's or Bonferroni's post hoc test, as appropriate. All *t*-test *t* and *p*-values and ANOVA *F* and *p*-values are reported in the legends to the Supplementary Figures; significant results of the contrast tests are further indicated by asterisks in the graphs. All statistical tests were two-tailed and $p < 0.05$ was considered statistically significant. For the complete set of raw data, see the data source file.

**Reporting summary**. Further information on research design is available in the Nature Research Reporting Summary linked to this article.

## Data availability

Candidate ubiquitination sites on BECN1 were taken from the mUbiSiDa data base (originally at http://202.195.183.4:8000/mUbiSiDa.php, now part of omicX at https://omictools.com/mubisida-tool). The source data underlying Figs. 1, 2, 4–9, and Supplementary Figs. 1–10 are provided as a Source Data file. All other data is available from the authors upon request.

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

## Acknowledgements

We are indebted to Mikko Taipale and Susan Lindquist for sharing the results of their FKBP51 interaction screen before publication. We thank Nicolas Heinemann, Anja Richter (both Charité), Artem Siemens, Maximilian Tschischka (both University of Bonn Medical Center) for excellent technical assistance. This work was partially funded by a NARSAD Young Investigator Award by Brain and Behavior Research Foundation, honoured by P&S Fund (to NCG, Grant ID 25348). NCG was further supported by a European Research Council starting grant (# 281338, GxE molmech, awarded to Elisabeth Binder) within the FP7 framework. CD was supported by the German Research Council (DFG) grant DFG SPP1596 (DR 772/10–2) and SFB-TR84 (TRR 84/3, A07), the Federal Ministry of Education and Research (BMBF) grant RAPID (#01KI1723A) and the EU Horizon 2020 grants COMPARE (# 643476) and EVAg (#653316). MAM was supported by the VW Foundation (#93345)." The SKP2 mutant plasmids were a kind gift from Wenyi Wei.

## Author contributions

N.C.G., V.M.C., C.D., M.A.M., T.R. designed and conceived the work. N.C.G., D.N., D.M., V.M.C., S.M., A.G., K.H., J.P., K.M., A.Z., A.S.Z., A.H., F.H., R.B.-W., M.B., B.M.-M., carried out experiments. N.C.G., D.N., D.M., V.M.C., S.M., A.G., K.H., J.P., K.M., A.Z., A.S.Z., A.H., F.H., R.B.-W., M.B., B.M.-M., C.D., M.A.M., T.R. analysed data. N.C.G., C.D., M.A.M., T.R. wrote the main paper text. N.C.G., prepared all figures. All authors reviewed the paper.

## Competing interests

The authors declare no competing interests.
