## [Peer Review File · Nature Communications]

Reviewers' Comments:

Reviewer #1:

Remarks to the Author:

The report by Gausse et al details the association between SKP2 and autophagy and how these pathways effect MERS-CoV replication. In this report, comprised of 2 large connected sets of data, SKP2 and its association with BECN1 is examined for their roles in regulating autophagy. The authors identify how SKP2 effects BECN1 levels via ubiquitination and how inhibition of SKP2 by either siRNA or inhibitory compounds effect BECN1 and autophagy. The report then presents data on the role of autophagy and MERS-CoV where live virus and individual MERS-CoV proteins are shown to effect autophagy flux. The large amount of data is technically impressive however significant questions arose on this review.

Major Issues

1. Throughout the report, the panels in the figures do not correlate with the references in the text. While the figure # is correct, the letter they relate to in the text is incorrect especially in all of the MERS-CoV figures.

2. MERS-CoV replication assays are reported as effects on mRNA levels. If autophagy is effecting MERS-CoV replication, the amount of virus released from cells is what should also be quantified by virus titer, not only viral mRNA levels. All experiments are performed at very low MOI for 48 hours suggesting amplification of virus propagation is effected however that data is not demonstrated in any figure. In high MOI infections where 1 cycle of virus replication is examined, is there an effect of autophagy on MERS-CoV replication? Both mRNA production and virus output? This data should accompany the figures where MERS-CoV replication is assessed.

3. The control protein for Figure 2 is TRAF6 to show that there is no effect on BECN1 protein levels via an additional ubiquitination protein. However TRAF6 is not included in the pulse chase experiments in Figure 2E and F. It should be shown as a comparison to vector alone.

4. The role of ubiquitination and the proteins that regulate it are the focus of Figure 3. In Figure 3 the control used in figure 3I is 3-MA treatment. First, how much 3-MA was used? Specificity of its inhibition diminishes at higher concentrations above 10uM. The data presented show that 3-MA is not working in this assay, as there is no decrease in long-term protein degradation observed unlike siSKP2 which shows an increase in long-term protein levels when knocked down. Why should 3-MA not show an increase in this experiment? This is the same result in Figure 4D where 3-MA has no effect on protein levels. This is not discussed in the text.

5. For Figure 4, there are several controls missing in the figure. First, there are no toxicity controls showing that the range of each inhibitor used is below significant cell toxicity levels. Secondly, the control in 4D using HBSS as an inducer of autophagy is not shown. Please show experimental evidence that in this experiment HBSS induces autophagy. Third, in 4E, BECN1 levels increase with all drugs; however LC3B increases with only 2 of them, and there doesn't seem to be a change in P62. There should be some correlation between these results but there is no consistency.

6. As MERS-CoV infection and proteins are considered for their role in autophagy, there are several conflicting results. First, the figure numbering does not match the text. Secondly, in Figure 5E, for the WCE blot it is not clear how you can observe both SNAP29 and STX17 in the same blot where it does not show up in the IP blots. Shouldn't they be in the same complex since they are crosslinked so they show as 1 band? The order of 5F and G should be reversed. And 5F is not clear. I assume it is a quantitation of 5G, however it does not match the data in 5G. The relationship between ATG14 monomeric and oligomeric forms and autophagy is not made clear. Are there published data showing oligomeric ATG14 actively inhibits autophagosome to lysosome fusion, or simply that it fails to promote fusion? Also, if MERS is inhibiting oligomerization, I would expect there to be an increase in monomer, which is not seen. Does total ATG14 amount change? Please clarify.

7. The MERS-CoV infections experiments in the supplemental figure S3 is also confusing. First, the cropped cells in S3E do not line up, please clean up the cropping. Secondly, there does not seem to be an increase yellow staining and a correlated decrease in red staining in the single cell shown in each panel. This example cell does not correlate with F. Third, the mutant viruses with either 4b or 5 deleted are great, however does the 4b deletion virus still contain the entire 4a coding region or was is 4a ORF effected when 4b was deleted? The oligos are in the methods section but no explanation in the text to the sites of the deletion. There are also no growth curves showing the 2 deletion viruses compared to WT virus. Do they grow equally well in the experiments or are they

attenuated in their growth under control conditions? Fourth, the virus output that is examined in K is only looking at mRNA with no statistically significant difference noted however it is stated in the text that the SKP2i inhibitor effects MERS growth. The K panel seems to be missing data. And the correlation should be with virus titer not mRNA since the autophagy effect would be expected to be on virus egress not transcription?

8. In the model figure and in the discussion, there is no description of data presented to show the direct effect of SKP2 on ATG14 oligomerization. Since this is a key feature of the model, it would be important to show that SKP2 siRNA or inhibition by drugs effects the ATG14 oligo/monomer relationship.

Minor Issues

1. Figure 1B. The whole cell lysates shown in the 2nd and 3rd lane for BECN1 (3rd blot down) are different however they are the same in the IP blot for BECN1 (2nd blot down). Please correct or explain why there are differences.

Reviewer #2:

Remarks to the Author:

Gassen et al reported here a very complete study of a novel regulation of autophagy by SKP2, a E3 ubiquitin ligase. They nicely demonstrated that SKP2 targets BECN1, an autophagy protein which is involved in autophagosome formation and maturation in two different complexes. SKP2 executes K48-ubiquitination at K402 of BECN1, allowing its proteosomal degradation. Activity of SKP2 is negatively controlled by FKBP51, which regulates SKP2 phosphorylation. Pharmacological and genetic inhibition of SKP2 leads to autophagy activation, with increased BECN1 levels, and increased interaction between lysosomal and autophagy SNARE proteins. They identified and studied a pharmacological compound, SMIP004, which strongly inhibits SKP2. The results presented here are novel, technically sound and contribute to a better understanding of autophagy regulation.

A second part of the article is dedicated to a possible therapeutic application of their data on Middle East respiratory syndrome (MERS) coronavirus. MERS-CoV is a human betacoronavirus responsible of a severe viral respiratory disease, first identified in Saudi Arabia in 2012. A third of reported patients with MERS have died and no antiviral treatment is currently available. They first demonstrated that MERS CoV blocks autophagy, in particular the fusion between autophagosome and lysosome, using several reliable autophagy monitoring experiments. They identified three non-structural viral proteins which can recapitulate the block of the autophagic flux observed during viral infection. They constructed recombinant viruses deleted for these genes but unfortunately poorly characterized their phenotype.

Finally they tested the effect of this SKP2 inhibitor, SMIP004 at 10 μ M, on MERS-CoV infection and observed an important inhibition of its multiplication. Unfortunately, they only evaluated viral genome production by RT-PCR at two different times post infection. Virus titers were not determined by plaque assay or TCID50 and no growth kinetics are presented, as expected for this kind of studies. Because a similar inhibition was observed with other compounds which interact with BECN1 and stimulate autophagy, they conclude that SMIP004 acts by stabilizing BECN1. They also tested FDA approved compounds reported to inhibit SKP2 and identified for several of them an antiviral activity against MERS-CoV. They conclude from this part of the work that inhibition of SKP2 leads to an important restriction of MERS CoV multiplication and represents an interesting therapeutic approach. This second part is quite interesting but suffers from speculative interpretations and lack of data from the literature. Some important experiments regarding the mechanism of antiviral activity of SMIP004 are missing and should be added to support their interpretation of the data.

In conclusion, this article reports totally new and important data on both autophagy regulation by SKP2 and on interaction between MERS-CoV and autophagy and a possible antiviral role of autophagy, which can be used as an innovative therapeutic approach.

Specific remarks:

Page 4

"CoV propagation and immune evasion is highly dependent on the formation of convoluted

membranes and double membrane vesicles (DMVs) that are reminiscent of autophagosomes (ref 17) pointing to a potential role of autophagy during the CoV life cycle.”

In the introduction the authors improperly and incompletely reported data from the literature regarding impact of autophagy on coronavirus infection.

Indeed, the presence of DMVs during Mouse Hepatitis virus infection (another betacoronavirus) has been reported in 2002 in the reference 17 cited by the authors. However, a more recent study, Regiorri et al demonstrated that DMVs are coated with the nonlipidated form of LC3 (LC3-I) and that LC3-II the autophagic marker and autophagy have no role on viral production. This was confirmed in the journal *Autophagy* by Zhao et al who demonstrated that MHV replication does not require Atg5. Therefore, it is established that autophagy has no role on MHV replication. However, it has been reported that autophagy negatively regulates TGEV replication (*Sci Rep* 2016). But another study reported a proviral role of autophagy on TGEV (*Oncotarget* 2016). Indeed, TGEV infection induces mitophagy to promote cell survival and possibly viral infection.

Page 11

“These results points to a block of autophagic flux by MERS-CoV infection at both early (BECN1) and late steps in the pathway”

Whereas data clearly demonstrated that MERS-CoV infection blocks autophagic flux and leads to an accumulation of immature autophagosomes, no evidence demonstrates that early steps of autophagy are reduced by MERS-CoV infection. A decrease of BECN1 expression is not sufficient to conclude this, when in parallel they observed an increase of the total number of autophagic vesicles during infection compared to control cells (Fig S3F). In the article (Cottam et al 2014, ref 41), the authors concluded that the non-structural protein nsp6 of IBV, an avian coronavirus, restricts autophagosome expansion by immunofluorescence by measuring the diameter of the autophagosomes.

The authors need therefore to modify this sentence.

Page 14

The results presented in Figure 5 clearly demonstrated that MERS-CoV blocks autophagosome maturation rather than initial steps of autophagosome formation. The titer of Figure 5 is therefore not adapted to the results and should be replaced by “blocks autophagic flux”.

Page 15 and Figure 6 panels A and B

“In order to confirm the influence of p4b and p5 on virus replication, we performed loss of...for the step-wise construction of an ORF4b- and ORF5-deleted MERS-CoV”

“The deletion of p4b or p5 resulted in 10-fold lower replication efficiency 48 hours p.i. (Fig. 6B)”

The characterization of the recombinant viruses presented in this study is clearly insufficient. No genome sequence analysis was reported. Growth kinetics of the different viruses showing viral titers (pfu/ml) at different times need to be added instead of measuring an inhibitory effect of the deletion (fig 6B).

More importantly, the authors did not report that in two previous independent studies (ref 44 of this article and Almazan et al, *Mbio*, 2013), construction of equivalent mutant MERS-CoV (no expression of orf5 or orf4ab) led to different results. Indeed, recombinant viruses lacking orf5 expression replicated to high titers (44). Similarly, Almazan et al observed that virus titers, cpe and plaque morphology of MERS-delta5 are similar to that of parental virus. In the case of MERS-delta4ab the viral titer was only 10 fold lower at 72hpi. How the authors explain this discrepancy? This can be related to an improper way to quantify viral production. Usually viral titers are determined by plaque assay (pfu/ml) in growth kinetics at different times of infection. In this study, they use RT-PCR in supernatant to detect viral genome but this experiment does not measure viral infectivity.

Figure 6 panel C

The authors concluded that mutant viruses do not control autophagy compared to parental virus. However, no quantification of LC3-II/LC3-I is available and the expression level of LC3-I seems important compared to other WB of the article.

Panel B and C: How is the multiplication of these mutant viruses in autophagy deficient cells? If growth kinetics of mutant viruses are similar to that of parental virus, this would allow the authors to confirm the antiviral role of autophagy on MERS-CoV and to reveal a possible requirement of orf4b or orf5 to counteract it.

Figure 6

The titer of the legend does not reflect the results. No data in this figure demonstrates that SKP2

inhibition reduces viral replication.

Page 15

Regarding SKP2 inhibitor, effect on viral replication is impressive, but same remark than above, growth kinetics and determination of viral titers are necessary. Moreover, to prove that the compound is acting via autophagy, effect of SMIP004 should be tested in autophagy deficient cells. This kind of approach was used for antibacterial activity of tat-beclin1 for example, in reference 46, Shoji-Kawata et al Nature 2013.

Page 17

ABT-737 and Tat-Beclin both inhibit MERS-CoV replication and they "leave the levels of BECN1 unchanged". Therefore, the authors cannot conclude just after that "increasing BECN1 is the relevant mechanism". We can make the same remark on page 18, on the mechanism of SKP2 inhibitor by enhancing BECN1 levels. Effects of overexpression of BECN1 on CoV replication should be tested. A siRNA against BECN1 will confirm the involvement of BECN1 and autophagy in CoV replication.

Page 18

The authors need to report that several of their tested drugs have well-described activity on autophagy, such as niclosamide, valinomycin, salinomycin and gossypol. They only confirmed previous studies. Moreover, these compounds are not only SKP2 inhibitors and specific autophagy inhibitors but have other activities since for example Niclosamide increases mitochondrial fission and valinomycin is a potassium ionophore. Once again, viral replication should be explored using plaque assay and growth kinetics.

Figure 7 panels B and C

The unusable way to determine the antiviral activity does not valorize the impressive results. Even if all these drugs inhibit SKP2, they have other activities. Therefore, the title needs to be modified unless impact of SKP2 is tested by gene silencing.

Discussion page 21 line 5

Same remark than above, unless they are testing SKP2 knock down, the authors cannot conclude on impact of SKP2 inhibition on viral multiplication. The authors should report in the discussion that autophagy has no antiviral effect on MHV, another betacoronavirus.

In conclusion regarding the antiviral activities the authors clearly identified components able to block MERS-CoV replication but the mechanism by SKP2 inhibition is not so far demonstrated.

Minor remarks:

Figure 1

"(C-F) FKP51.... Transfected with vector control... or FKP51" add (Co) after control

Figure 3 panel I

Information regarding \$\$ is missing in the legend. Unfortunate that only LLP degradation was tested with siRNA against SKP2 and no other readout for autophagy such as LC3 of p62.

Page 10

"Targeting SKP2 INCREASED protein degradation"

Page 11

"we aimed to test if blocking the SKP2 dependent autophagy inhibition influences CoV replication". To improve the understanding of this sentence, it is better to say "if stimulation of autophagy by SKP2 inhibitor influences MERS-CoV replication. Add MERS instead of just CoV because, it was tested only for this coronavirus.

Page 13-15

Several mistakes in the labelling of panels for figure S3

Page 13 S3DE is S3EF; S3F is in fact S3G, S3G is S3H

page 15 S3H is S3I I is J and J is K

Moreover in Figure S3K what is the condition "Co"? sam question for Figure S5 panel I. where are the white histograms?

pages 2 4 15 17 20 22

The word "propagation" is used several times in the text, but corresponds in fact to viral cell-to-cell spread and is not tested here. It should be replaced by viral multiplication or production.

Finally the title does not totally reflect data presented in the article, because impact of SKP2 on viral replication has been explored only using compounds which have other activities than just inhibiting SKP2.

Reviewer #3:

Remarks to the Author:

Autophagy is a cellular homeostatic pathway that can mediate degradation of obsolete organelles, misfolded proteins and other cargo like viral components. Being a destructive process, autophagy has to be tightly controlled. Indeed, intricate signalling pathways regulate both induction and flux rates of autophagy. Recent reports have highlighted the impact of autophagy on many cellular processes and its role as an anti-viral mechanism.

Gassen et al had previously identified FKBP51 as a beclin-binding protein, which mediates induction of autophagy (Gassen et al, 2014&2015). Continuing their work on autophagy, they elaborate on the mechanism. FKBP51 is assembling an autophagy-promoting complex, which deactivates/dephosphorylates the E3 ligase SKP2. Inactivated SKP2 no longer mediates K48-linked polyubiquitination of beclin-1 thereby stabilizing its protein levels and promoting autophagy. Furthermore, the authors characterise the interplay between autophagy and MERS-CoV. While infection inhibits autophagic flux via several viral proteins, exogenous activation of autophagy by pharmacological inhibition of SKP2 combats MERS-CoV infection.

Experiments characterising autophagy are conducted in a convincing way and the data supports the claims of the authors. The paper by Gassen et al features an elegant exploration of the complicated mechanism of FKBP51 and links SKP2 to beclin-1. They manage to assemble puzzle pieces from previously published studies about the regulation of SKP2 (e.g. Rodier et al, 2008 and others) complemented by their own data to suggest assembly of a multi-protein complex that regulates stability of beclin-1. Finally, this study provides novel data on how to combat viral infections (here MERS-CoV) using insights gained from understanding the molecular mechanism of autophagy induction and virus-autophagy interplay. However, occasionally it feels like there are two (interesting) separate stories existing side-by-side (Mechanism of FKBP51/SKP2, Interplay between autophagy and MERS-CoV), which need to be tied together more tightly.

A few specific issues need to be addressed as well:

Major:

1. Treatment with drugs can have multiple effects on viral replication. Therefore, even though all used drugs presumably target a similar pathway, it has to be made sure that autophagy is responsible for viral attenuation. The authors should conduct experiments in which autophagy is induced (e.g. by mentioned drugs), but the effects of this induction are blocked. For example, can the growth attenuation be rescued if autophagic turnover is blocked by chloroquine, or LC3B conjugation inhibited by ATG5 KO/KD? Furthermore, the chosen drugs used to inhibit SKP2 (even though it is backed up by literature) need direct evidence that SKP2-dependent beclin-1 ubiquitination is reduced.

2. The interesting data on MERS-CoV prompts the question, how do other viruses, which are known to be sensitive towards autophagy, react towards inhibition of SKP2. Experiments exploring the effect of the drugs on e.g. Sindbis Virus or HIV-1 replication should clarify that issue and would complement the MERS-CoV data and make it less focused on treating a single virus with a system potentially applicable for other viral pathogens.

3. Fig. 2C: To show that SKP2 is the E3 ligase which ubiquitinates beclin-1, the authors should include a SKP2 E3-ligase null mutant and monitor beclin-1 levels/degradation. The effect observed in Fig. 2C-F should be rescued by MG132 treatment, to prove its proteasome dependent degradation.

4. Fig. 3: A few additional details have to be clarified: Is Beclin-1 K402R resistant towards degradation induced by either PHLPPi treatment or SKP2 overexpression? Since AKT1 is involved in the cascade, does pharmacological inhibition of AKT1 lead to stabilization of beclin-1?

Minor:

1. The abstract needs rephrasing to emphasise the achievements of this paper in a more cohesive and appealing way. The current abstract undersells the story.

2. The title of the paper could be shortened to sound less convoluted.

3. Writing of the results section (first part) might need improvement to make it more accessible. Furthermore, mechanistic details could be presented more cohesively to focus on the achievements of the paper.

4. Please add size markers to all the western blots.
5. Fig. 1 B: Stabilization of beclin-1 levels by MG132 has to be compared to NH₄Cl treatment, to rule out turnover by autophagy.
6. Fig. 2A,B: It would be interesting to see the whole Western Blots (especially for USP18 and USP36, including the input).
7. Fig. 2C,D: Since SKP2 is regulated by phosphorylation. It would be curious to see how phosphomimetic and phospho-null mutants of SKP2 would impact beclin-1 stability.
8. Fig. 3 I: Please show one exemplary western blot of a protein degraded by autophagy.
9. Fig. 4 E: Please add a quantification for the western blots as shown in the figure.
10. Fig. 5D and E: Both figure panels could be moved to the supplements, as the effect observed is only marginal and other data presented is stronger.
11. Fig. 6A: While interesting, the sketch could be moved to the supplements.
12. Fig. 6G: Could be moved to the supplements
13. Fig. 6E: While manual counting is certainly being used a lot, these assays could be alternatively quantified using flow cytometry to get data, which might be less biased by manual observation.
14. Fig. 7 A,B: Is there a correlation between autophagy induction and MERS-CoV inhibition ?
15. Fig. 7 B: For the viral inhibition assays, it would be beneficial to show in the supplements the actual copy numbers of viral genome. To get an idea of the magnitude of the effect viral titres for one assay have to be determined showing the effect of autophagy induction on MERS-CoV.
16. Fig. 7C: Is it fold inhibition?
17. Fig. 7D: Might be better as a graphical abstract, not as a figure subpanel.
18. Page 5: Please add a short sentence with the reasoning why only 'SKP2, USP18 and USP36 have the potential to influence K48-linked' etc.
19. Page 11: mRFP does not resist degradation in the lysosome. It is merely not inactivated by low pH like GFP.
20. The authors should discuss the effects of the different viral proteins of MERS-CoV on autophagy, and possible implications in the replication.

Point by point rebuttal to the reviewers' comments

Reviewer #1

We thank reviewer #1 for the insightful, critical and constructive comments. We performed new experiments to address these comments, and moderated our conclusions as suggested where necessary, while balancing the modifications keeping comments of reviewer #3 in mind who suggested to better highlight the merits of the study.

Major Issues

Comment 1. "Throughout the report, the panels in the figures do not correlate with the references in the text. While the figure # is correct, the letter they relate to in the text is incorrect especially in all of the MERS-CoV figures."

Response:

We apologize for these errors; we double-checked to make sure all the numbers and letters in the text correlate to the right figure and panel.

Comment 2. "MERS-CoV replication assays are reported as effects on mRNA levels. If autophagy is effecting MERS-CoV replication, the amount of virus released from cells is what should also be quantified by virus titer, not only viral mRNA levels. All experiments are performed at very low MOI for 48 hours suggesting amplification of virus propagation is effected however that data is not demonstrated in any figure. In high MOI infections where 1 cycle of virus replication is examined, is there an effect of autophagy on MERS-CoV replication? Both mRNA production and virus output? This data should accompany the figures where MERS-CoV replication is assessed."

Response:

We agree with the reviewer. In order to directly show that virus production (on request of reviewer 2 we are avoiding the word "propagation", because this refers to spreading from cell to cell which we do not show directly) is affected we determined the reduction of plaque forming units/ml in parallel to virus RNA copies per ml. Based on this comment and comments from the other reviewers, we performed new experiments and now display the fold difference (log10 scale) of the RNA analysis (genome equivalents, GE) and the plaque forming units, as well as the raw data of the genome equivalents in the new Figures 9B and S9A-C. Figure S9A shows the unprocessed data of the detected genome equivalents per ml, which are in good agreement with the GE data of the first submission. Clearly, the RNA analysis and the determination of the pfus revealed the same pattern of activity, indicating that both assays are valid to investigate the influence of autophagy inducers on MERS-CoV replication/growth.

New figure panel 9B:

Legend to Figure 9B: "VeroB4 cells were infected with MERS-CoV (MOI = 0.001), treated with the indicated drugs, and MERS-CoV genome copies were determined by RT-PCR 24 h and 48 h p.i.; data are presented as (log₁₀) fold difference in relation to the DMSO control. Unprocessed data and pfu results are presented in Fig. S9A-C."

New figure panels S9A-C:

Legend to Figure 9A-C: "VeroB4 cells were infected with MERS-CoV (MOI = 0.001), treated with the indicated drugs, and MERS-CoV genome copies (A) as well as PFUs (B,C) were determined 24 h and 48 h p.i.; data represent one of two independent experiments showing the mean values of biological triplicates and are displayed as

log10 GE/mL (A) (fold difference in Fig. 9B) and fold difference of the pfus (B) derived from the data presented in panel C.”

We chose to examine the role of autophagy on MERS-CoV growth in multi-cycle replication assays in order to identify meaningful effects that reflect physiological conditions. Only a few MERS-CoV particles are necessary to establish an infection in the respiratory tract. All assays were, therefore, performed with a low MOI. Importantly, we used two different time points (e.g. 24 and 48 hours post infection) to account for effects during the exponential growth of the virus (24 hours) and the plateau of virus production (48 hours).

Comment 3. The control protein for Figure 2 is TRAF6 to show that there is no effect on BECN1 protein levels via an additional ubiquitination protein. However TRAF6 is not included in the pulse chase experiments in Figure 2E and F. It should be shown as a comparison to vector alone.

Response:

We performed the requested experiment and replaced panels E and F of Figure 2. On the request of reviewer #3, these panels now also display conditions including the proteasome inhibitor MG132. Similarly to the cycloheximide assay (panels C and D), TRAF6 did not affect protein stability of Beclin1.

New Figure panels 2E and F (changed to respect the concerns of reviewers 1 and 3):

Legend of Figure 2E,F: “The conditions of C,D were also used in the pulse-chase assay, performed as in Fig. 1E,F, to determine BECN1 stability.”

Comment 4. “The role of ubiquitination and the proteins that regulate it are the focus of Figure 3. In Figure 3 the control used in figure 3I is 3-MA treatment. First, how much 3-MA was used? Specificity of its inhibition diminishes at higher concentrations above 10uM. The data presented show that 3-MA is not working in this assay, as there is no decrease in long-term protein degradation observed unlike siSKP2 which shows an increase in long-term protein levels when knocked down. Why should 3-MA

not show an increase in this experiment? This is the same result in Figure 4D where 3-MA has no effect on protein levels. This is not discussed in the text.”

Response:

3-MA was used at a concentration of 10 mM, which is the concentration typically used for the inhibition of autophagy (for example Cell 2010; 140: 313–326). We are aware of the reports that 3-MA can have effects beyond blocking autophagy, and even promote autophagy under certain conditions. This is also stated in the third edition of the "Guidelines for the use and interpretation of assays for monitoring autophagy" (AUTOPHAGY 2016; 12: 1–222), page 54: "For example, 3-MA is commonly used to inhibit starvation- or rapamycin-induced autophagy, but it has no effect on BECN1-independent forms of autophagy, and some data indicate that this compound can also have stimulatory effects on autophagy". In our case, the goal was to investigate BECN1-dependent autophagy, for which 3-MA should work according to these guidelines. Moreover, we do see inhibition of induced autophagy, because the enhanced proteolysis of long-lived proteins (induced by si-RNA targeting Skp2 that enhances BECN1 protein levels, Fig. 2G) are abolished by 3-MA (Figure 4M, former Figure 3I). Thus, 3-MA worked in its "typical way", i.e. blocking induced autophagy, as indicated by decreased proteolysis of long-lived proteins (meaning increased protein levels, not a decrease). The guidelines further state on page 186 that "At concentrations >10 mM 3-MA inhibits other kinases such as AKT (Ser473), MAPK/p38 (Thr180/Tyr182) and MAPK/JNK (Thr183/Tyr185)." We were not exceeding 10 mM in our experiments. At lower concentrations, the inhibition of autophagy may not be complete (Molecular and Cellular Neuroscience 1999; 14: 180–198. Proc. Nat. Acad. Sci. USA 1982; 79: 1889-1892).

Also in Figure 5D (former figure 4D), we believe that 3-MA displays no unusual effects. It does not change basal autophagy, but blocks the induction (as indicated by the increased proteolysis of long-lived proteins) of autophagy by all compounds that induce autophagy. An effect of 3-MA on induced, but not basal autophagy has been observed before (for example Fig. S1 of J Pharmacol Exp Ther. 2010; 333: 454–464).

We changed the manuscript by providing the concentration of 3-MA in the figure legends as well as in the methods section, by correcting the description of the turn-over assay in the results section, and by providing more citations. We noted that there was a wrong wording in our description of the turn-over assay, which may have misled this reviewer, and we apologize for this. The section now reads (Figure 3I is now Figure 4M):

"In order to verify that reduced SKP2 activity drives autophagy in our cellular set-up, we used siRNA and determined the degradation of long-lived proteins⁴⁸. Targeting SKP2 increased proteolysis, but not in the presence of the autophagy inhibitor 3-methyladenine (3-MA, Fig. 4M). Furthermore, we observed decreased levels of p62 (Fig. S1G), which typically goes along with enhanced autophagy⁴⁸. All these data are consistent with an inhibitory effect of SKP2 on autophagy regulated by a phosphorylation cascade involving FKBP51, PHLPP and AKT1.

Comment 5. "For Figure 4, there are several controls missing in the figure. First, there are no toxicity controls showing that the range of each inhibitor used is below significant cell toxicity levels. Secondly, the control in 4D using HBSS as an inducer of autophagy is not shown. Please show experimental evidence that in this experiment HBSS induces autophagy. Third, in 4E, BECN1 levels increase with all drugs; however LC3B increases with only 2 of them, and there doesn't seem to be a change in P62. There should be some correlation between these results but there is no consistency."

Response:

To 5.1: Figure S2A displays the toxicity controls and shows that each inhibitor is used at a concentration below significant cell toxicity.

To 5.2: We performed flux assays with HBSS and BafA1 which confirmed the induction of autophagic flux by HBSS. These data are now presented as Figure S2D:

Legend to figure S2D: HBSS control in VeroB4 cells to demonstrate the induction of autophagic flux. Representative Western blots are displayed.

To 5.3: The quantification of Figure 4E (now figure 5E) is provided in Fig. S2E-G (former panels B-C). Actually, there is at least some correlation. Compound C1 shows no significant effects on BECN1 (non-significant increase), LC3B-I/II, and p62. SMER3 displays a trend induction for BECN1, a significant effect on LC3B-II/I ($p < 0.05$) and a trend for p62, while SMIP004 displays significant effects for all three of these parameters. Sometime, it is not possible to capture the average for all measurements in one example blot, so for quantitative considerations, Fig. S2E-G should be consulted.

We changed the section of the results describing Figure 5E, S2B-G (end of page 12): "Furthermore, SMIP004 most profoundly affected not only BECN1, but also LC3B-II/I and P62 (example blot in Fig. 5E, quantifications in Fig. S2E-G) that are established markers of autophagy."

Comment 6. "As MERS-CoV infection and proteins are considered for their role in autophagy, there are several conflicting results. First, the figure numbering does not match the text. Secondly, in Figure 5E, for the WCE blot it is not clear how you can observe both SNAP29 and STX17 in the same blot where it does not show up in the IP blots. Shouldn't they be in the same complex since they are crosslinked so they show as 1 band? The order of 5F and G should be reversed. And 5F is not clear. I assume it is a quantitation of 5G, however it does not match the data in 5G. The relationship between ATG14 monomeric and oligomeric forms and autophagy is not made clear. Are there published data showing oligomeric ATG14 actively inhibits autophagosome to lysosome fusion, or simply that it fails to promote fusion? Also, if MERS is inhibiting oligomerization, I would expect there to be an increase in monomer, which is not seen. Does total ATG14 amount change? Please clarify."

Response:

To 6.1: We apologize for the errors in Figure numbering, which we corrected.

To 6.2 Figure 5E: WCE in Figure 5E (now Fig. 6H) does not represent cross-linked material. It rather is a control to show the non-cross-linked proteins. Thus, the protein bands of the WCE represent only the designated protein. After cross-linking and IP of STX17, proteins cross-linked to STX17 (as well as STX17 itself) were detected individually at the combined molecular weight (i.e. STX17 + VAMP8 and STX17 + SNAP29). This now is described in the figure legend as follows: "(H) MERS-CoV affects SNARE protein interactions. VeroB4 cells were infected with MERS-CoV (MOI = 0.001), cross-linked with disuccinimidyl suberate (DSS, 75 μ M) 48 h p.i. for 30 min and harvested. The SNARE complex protein STX17 was immunoprecipitated and the eluate was probed for interacting VAMP8 and SNAP29 by western blotting (the quantification represents the bands detected at the combined molecular weights of the cross-linked proteins, i.e. STX17 + VAMP8 and STX17 + SNAP29). WCE = whole cell extract of vehicle exposed cells, i.e. no cross-linking."

To 6.2 Figure 5F,G: We reversed Figures 5F and 5G, as requested (they are now Figs. 6E,F). We argue that the example blot is well chosen in this case, because it reflects the quantification fairly well. MERS-CoV infection does not change the amount of monomeric ATG14 (black bars, lower bands), but strongly decreases the amount of oligomeric ATG14 (grey bars, upper bands). We completely agree with the reviewer that the observation of strongly decreased ATG14 oligomerisation without apparent increase in ATG14 monomers is difficult to explain as we cannot come up with an immediately obvious reason. Likewise, monomers did not change in the face of the increase in oligomers upon inhibition of SKP2 (Fig. 8D,E, former 6F and figureS2H). When looking at the conditions in the absence of cross-linker, total ATG14 was not changed (Compare lanes 1 and 3 in figures 6F [former 5G] and 8D [former 6F]). Nevertheless, changes in ATG14 oligomerisation concomitant with unchanged ATG14 monomers was also reported in the original Nature paper that introduced the role of ATG14 oligomerisation in autophagy (Figure 3b and extended data Figure 5k, Nature 2015, 520: 563-566). We cannot exclude the possibility that cross-linking stabilizes ATG14, but this is very speculative at this point. In any case, the published role of ATG14 oligomerisation was the basis for this part of our experiments. It promotes membrane tethering and fusion of autophagosomes to

endolysosomes (*Nature* 2015, 520: 563-566). We mention this when we describe the rationale of our approach. It is first mentioned in the description of the new figure S2H (page 14):

"In addition, it has been shown that ATG14 oligomerizes to dimers and tetramers, which is essential for autophagy by promoting STX17 binding and autophagosome-lysosome fusion⁷. Using capillary-based electrophoresis allowing for better resolution at high molecular weight we observed ATG14 oligomerization to 7-8mers, which was enhanced by SMIP004 (Fig. S2H, I)."

Comment 7. "The MERS-CoV infections experiments in the supplemental figure S3 is also confusing. First, the cropped cells in S3E do not line up, please clean up the cropping. Secondly, there does not seem to be an increase yellow staining and a correlated decrease in red staining in the single cell shown in each panel. This example cell does not correlate with F. Third, the mutant viruses with either 4b or 5 deleted are great, however does the 4b deletion virus still contain the entire 4a coding region or was is 4a ORF effected when 4b was deleted? The oligos are in the methods section but no explanation in the text to the sites of the deletion. There are also no growth curves showing the 2 deletion viruses compared to WT virus. Do they grow equally well in the experiments or are they attenuated in their growth under control conditions? Fourth, the virus output that is examined in K is only looking at mRNA with no statistically significant different noted however it is stated in the text that the SKP2i inhibitor effects MERS growth. The K panel seems to be missing data. And the correlation should be with virus titer not mRNA since the autophagy effect would be expected to be on virus egress not transcription?"

Response:

To 7.1 and 7.2: We improved the example images of Figure S3E (now figure 6D) as requested:

We noted that in the large word files some figures come out a little blurry, even varying from computer screen to computer screen. We thus are submitting a separate file compiling all the figures for the main text and the supplement.

To 7.3: We now provide an overview picture of the mutations in the supplementary material (new Figure S4B). For MERS-del4b, the nucleotide positions 26182- 26751 were deleted. As shown in the picture, the complete orf4a is still present. For the MERS-del5 mutant nucleotide positions 26840-27515 were deleted. Therefore, the ORF5 (encompassing 675 nucleotides from start to stop codon) was completely deleted. The regulatory element of ORF5 (TRS-5) was retained to maintain an equal number of subgenomic mRNAs between wildtype and mutant virus.

New figure S4B:

Legend to Figure S4B: Overview picture of the cloning strategy for construction of the MERS-CoV mutations. For MERS-del4b, the nucleotide positions 26182- 26751 were deleted. As shown in the picture, the complete orf4a is still present. For the MERS-del5 mutant nucleotide positions 26840-27515 were deleted. Therefore, the ORF5 (encompassing 675 nucleotides from start to stop codon) was completely deleted. The regulatory element of ORF5 (TRS-5) was retained to maintain an equal number of subgenomic mRNAs.

We included the following passage in the Material and Methods section:

"Construction of a MERS-CoV cDNA clone lacking accessory ORFs: The accessory ORFs (ORF4b or ORF5) were deleted using a two-step markerless Red Recombination system as extensively described^{84, 92}. Briefly, in a first step a kanamycin resistance gene was amplified by an overhang extension PCR from the pEP-KanS vector (kindly provided by K. Tischer and K. Osterrieder) with primers containing homologous MERS-CoV recombination sites, the desired mutation and an I-SceI restriction site (MERS-delta4b-ABC-Kana-F: 5'-ATTCGCGCAAAGCGAGGAAGAGGAGCCATTCTCAACTA ATGATGTTGTCTCCATACGGTCTTTAGGGATAACAGGGTAATCGATTT-3' and MERS-delta 4b-BCD-Kana-R: 5'-TTGTTTATTACCCTGATTGGAAGACCGTATGGAGACAACATCATTAG TTGGAGAATGGCTCCTGCCAGTGTTACAACCAATTAACC-3'; MERS-delORF5_Kana-F: 5'-TTCTTATCCCATTTTACATCATCCAGGATTTAACGAAGTGCAGCTCTGCGCTACTATGGTAG GGATAACAGGGTAATCGATTT-3' and MERS-delORF5_Kana-R: 5'-GATTAGCCTCTAC ACGGG ACCCATAGTAGCGCAGAGCTGCAGTTCGTTAAATCCTGGATGCCAGTGTTACAAC CAATTAACC-3'). We would like to point out that this design preserves the regulatory sites of ORF5, in contrast to a previous approach⁹³. Our rationale was to preserve an equal number of subgenomic mRNAs. E. coli GS1783 cells (kindly provided by K. Tischer and K. Osterrieder) containing the MERS-CoV cDNA clone were transfected

with the PCR product in order to induce the recombination of the transfer construct into the MERS-CoV backbone. In a second step, the kanamycin selection marker was excised by an inducible *in vivo* I-SceI cleavage and the second Red recombination event. Rescue and stock production of the recombinant viruses were performed as described elsewhere⁸⁴.”

To the question whether these truncated viruses grow equally well: Figure S4E,F shows virus growth of the 3 different recombinant viruses in Vero B4 wt and ATG5 KO cells. ATG5 KO cell experiments were performed in response to the request of reviewers #2 and #3 and are included here to not overly expand the number of figures and panels. With respect to wt Vero B4 cells, the MERSdel5 and MERSdel4b viruses grow approximately 0.5-1 log₁₀ less than the MERS wt virus, as congruently revealed by assessing GEs and PFUs at 48h p.i.:

Legend to figure S4E,F: Deletion of ATG5 in the host cells VeroB4 does not abolish the difference in replication between mutant and wt MERS-CoV. VeroB4 wt or ATG5 KO cells were infected with wt or mutant recombinant MERS-CoV (rEMC) (MOI = 0.001) and genome copies (E) as well as plaque forming units (F) (PFU) were determined at 24 and 48 h p.i.. Unprocessed data are presented.

To 7.4: We agree that Figure S3K could have been confusing as we only showed the growth difference in relation to DMSO-treated cells, but did not display data of these vehicle-treated cells directly. In the revised manuscript we included a modified version showing fold change difference between DMSO vs SKPi-treated cells (Fig. 8A). In the supplement, we also included the unprocessed data showing the GE/ml (S5A). In Figure 9B,C we establish that GE/ml and PFU/ml can both be used to obtain an estimate of the virus growth.

Legend to Fig. 8A: The SKP2 inhibitor (SKP2i) efficiently inhibits MERS-CoV replication. VeroB4 cells were infected with MERS-CoV (MOI = 0.001) and treated with SKP2i (10 μM) or DMSO (Co = control). MERS-CoV GE were determined by real-time RT-PCR at 24 and 48 h p.i., data are presented as fold difference (black bars, SKP2i condition) in comparison to control (grey bars, Co = DMSO condition). Raw data presenting GE/ml is shown in Fig. S5A.

Legend to Fig. S5A: SKP2i reduces MERS-CoV replication. VeroB4 wt cells were infected with MERS-CoV (MOI = 0.001) and genome copies were determined at 24 and 48 h p.i.. Unprocessed data are presented from which the fold differences presented in Fig. 8A are derived.

Comment 8. "In the model figure and in the discussion, there is no description of data presented to show the direct effect of SKP2 on ATG14 oligomerization. Since this is a key feature of the model, it would be important to show that SKP2 siRNA or inhibition by drugs effects the ATG14 oligo/monomer relationship."

Response:

Figure 8D,E (former figure 6F) shows the effect of SKP2 inhibition on ATG14 oligomerisation in MERS-CoV infected cells. The inhibitor produced more ATG14 oligomers, but had no effect on the amount of ATG14 monomers. As mentioned in our response to major point 6, we find the absence of effect on ATG14 monomers in the presence of pronounced effects on ATG14 oligomers difficult to explain, but this has been observed by others in different settings as well. The idea is not that SKP2 causes ATG14 oligomerisation directly, but that this has recently been shown as crucial step in functional autophagy (Nature 2015 520: 563-566) and thus was included in our analyses.

We mention this now first in the results to new Figure S2H, as also explained in our response to comment 6.2: "In addition, it has been shown that ATG14 oligomerizes

to dimers and tetramers, which is essential for autophagy by promoting STX17 binding and autophagosome-lysosome fusion⁷. Using capillary-based electrophoresis allowing for better resolution at high molecular weight we observed ATG14 oligomerization to 7-8mers, which was enhanced by SMIP004 (Fig. S2H,I)”.

Further, the description of panel D in the legend of Figure 9 (former Figure 7) now is: “Summary scheme: SKP2 leads to K48-linked poly-ubiquitination and thus degradation of BECN1. The effect of SKP2 can be diminished in two ways, either by chemical inhibition or by FKBP51, which scaffolds protein interactions ultimately leading to SKP2 inactivation through inhibiting its phosphorylation. Both scenarios enhance autophagy, which involves ATG14 oligomerization (probably 7-8mers) as newly described essential step in functional autophagy⁷. MERS-CoV reduces autophagy through distinct viral proteins leading to blockade of autophagosome-lysosome fusion and ATG14 oligomerization. Compounds inhibiting SKP2 reinstate autophagy and efficiently reduce viral production.”

Minor Issues

Minor comment 1. “Figure 1B. The whole cell lysates shown in the 2nd and 3rd lane for BECN1 (3rd blot down) are different however they are the same in the IP blot for BECN1 (2nd blot down). Please correct or explain why there are differences.”

Response:

We replaced the blot with a more suitable example:

B

Reviewer #2

"Gassen et al reported here a very complete study of a novel regulation of autophagy by SKP2, a E3 ubiquitin ligase. They nicely demonstrated that SKP2 targets BECN1, an autophagy protein which is involved in autophagosome formation and maturation in two different complexes. SKP2 executes K48-ubiquitination at K402 of BECN1, allowing its proteosomal degradation. Activity of SKP2 is negatively controlled by FKBP51, which regulates SKP2 phosphorylation. Pharmacological and genetic inhibition of SKP2 leads to autophagy activation, with increased BECN1 levels, and increased interaction between lysosomal and autophagy SNARE proteins. They identified and studied a pharmacological compound, SMIP004, which strongly inhibits SKP2. The results presented here are novel, technically sound and contribute to a better understanding of autophagy regulation.

A second part of the article is dedicated to a possible therapeutic application of their data on Middle East respiratory syndrome (MERS) coronavirus. MERS-CoV is a human betacoronavirus responsible of a severe viral respiratory disease, first identified in Saudi Arabia in 2012. A third of reported patients with MERS have died and no antiviral treatment is currently available. They first demonstrated that MERS CoV blocks autophagy, in particular the fusion between autophagosome and lysosome, using several reliable autophagy monitoring experiments. They identified three non-structural viral proteins which can recapitulate the block of the autophagic flux observed during viral infection. They constructed recombinant viruses deleted for these genes but unfortunately poorly characterized their phenotype.

Finally they tested the effect of this SKP2 inhibitor, SMIP004 at 10 μ M, on MERS-CoV infection and observed an important inhibition of its multiplication. Unfortunately, they only evaluated viral genome production by RT-PCR at two different times post infection. Virus titers were not determined by plaque assay or TCID50 and no growth kinetics are presented, as expected for this kind of studies. Because a similar inhibition was observed with other compounds which interact with BECN1 and stimulate autophagy, they conclude that SMIP004 acts by stabilizing BECN1. They also tested FDA approved compounds reported to inhibit SKP2 and identified for several of them an antiviral activity against MERS-CoV. They conclude from this part of the work that inhibition of SKP2 leads to an important restriction of MERS CoV multiplication and represents an interesting therapeutic approach. This second part is quite interesting but suffers from speculative interpretations and lack of data from the literature. Some important experiments regarding the mechanism of antiviral activity of SMIP004 are missing and should be added to support their interpretation of the data. In conclusion, this article reports totally new and important data on both autophagy regulation by SKP2 and on interaction between MERS-CoV and autophagy and a possible antiviral role of autophagy, which can be used as an innovative therapeutic approach."

Response:

We thank the reviewer for the constructive comments. We believe to have addressed the major concerns of this reviewer by i) characterizing the phenotype of the MERS-

CoV deletion mutants (growth curves, Figs. S4E,F), ii) quantification of infectious (plaque forming) units in addition to the previous detection of genome equivalents (Figures 7A, S4F, S5B,C, S9B,C) and iii) including Vero ATG5 KO cells in addition to Vero wt cells showing that depletion of ATG5 substantially increases the production of MERS-CoV infectious particles (Figs.7A,B S4E,F). The detailed responses are provided below.

Specific remarks:

Comment Page 4

“CoV propagation and immune evasion is highly dependent on the formation of convoluted membranes and double membrane vesicles (DMVs) that are reminiscent of autophagosomes (ref 17) pointing to a potential role of autophagy during the CoV life cycle.

In the introduction the authors improperly and incompletely reported data from the literature regarding impact of autophagy on coronavirus infection.

Indeed, the presence of DMVs during Mouse Hepatitis virus infection (another betacoronavirus) has been reported in 2002 in the reference 17 cited by the authors. However, a more recent study, Reginori et al demonstrated that DMVs are coated with the nonlipidated form of LC3 (LC3-I) and that LC3-II the autophagic marker and autophagy have no role on viral production. This was confirmed in the journal Autophagy by Zhao et al who demonstrated that MHV replication does not require Atg5. Therefore, it is established that autophagy has no role on MHV replication. However, it has been reported that autophagy negatively regulates TGEV replication (Sci Rep 2016). But another study reported a proviral role of autophagy on TGEV (Oncotarget 2016). Indeed, TGEV infection induces mitophagy to promote cell survival and possibly viral infection.”

Response:

We are grateful for these supportive suggestions and included all suggested citations. It appears to us that no general role of autophagy in CoV replication has been found. The revised section in the introduction now reads as follows:

“.....A role of autophagy is suggested for another CoV, the mouse hepatitis virus (MHV), by the observation that multiplication of, and immune evasion by the virus is highly dependent on the formation of convoluted membranes and DMVs that are reminiscent of autophagosomes²⁷. However, the link of MHV replication to these DMVs did not appear to involve autophagy as the deletion of the pivotal autophagy genes ATG7 or ATG5 did not affect MHV replication^{28,29}. Of note, also the induction of autophagy by starvation did not significantly change MHV replication²⁹. On the other hand, results of an earlier study employing ATG5 knockout cells suggested that autophagy is required for the formation of DMV-bound MHV replication complexes thereby significantly enhancing the efficiency of viral replication¹⁸. Furthermore, pharmacological or genetic manipulation of autophagy showed that replication of another CoV, the Transmissible Gastroenteritis virus (TGEV), is negatively regulated by autophagy³⁰. In contrast, another study reported enhancement of TGEV

*replication by autophagy*³¹. Thus, no general role of autophagy in CoV replication could be established yet.”

Comment Page 11

““These results points to a block of autophagic flux by MERS-CoV infection at both early (BECN1) and late steps in the pathway”

Whereas data clearly demonstrated that MERS-CoV infection blocks autophagic flux and leads to an accumulation of immature autophagosomes, no evidence demonstrates that early steps of autophagy are reduced by MERS-CoV infection. A decrease of BECN1 expression is not sufficient to conclude this, when in parallel they observed an increase of the total number of autophagic vesicles during infection compared to control cells (Fig S3F). In the article (Cottam et al 2014, ref 41), the authors concluded that the non-structural protein nsp6 of IBV, an avian coronavirus, restricts autophagosome expansion by immunofluorescence by measuring the diameter of the autophagosomes.

The authors need therefore to modify this sentence.”

Response:

We modified and expanded the description as follows (üage 14, former Fig. S3F is now Fig. 6C):

“Infection with MERS-CoV further led to an increase of the total number of phagocytic vesicles (sum of AL + AP) per cell (Fig. 6C,D). However, the number of successfully formed AL was reduced significantly, indicating that AP can form but not fuse with lysosomes when cells are infected. A fusion block was also evidenced by the significantly reduced ATG14 oligomerization, essential for AP-lysosome fusion⁷, in infected cells (Fig. 6E,F) and by the increase of the autophagy target P62 (Fig. S3C).”

Comment Page 14

“The results presented in Figure 5 clearly demonstrated that MERS-CoV blocks autophagosome maturation rather that initial steps of autophagosome formation. The titer of Figure 5 is therefore not adapted to the results and should be replaced by “blocks autophagic flux”.”

Response:

We agree and modified the title of Figure 5 (now figure 6) which now reads: „MERS-CoV blocks autophagic flux“. Even though the figure has changed somewhat (new panels C,D, former panel D is now in the supplement), the suggested title still fits well, in our opinion.

Comment Page 15 and Figure 6 panels A and B

“In order to confirm the influence of p4b and p5 on virus replication, we performed loss of...for the step-wise construction of an ORF4b- and ORF5-deleted MERS-CoV”
“The deletion of p4b or p5 resulted in 10-fold lower replication efficiency 48 hours p.i. (Fig. 6B)”

The characterization of the recombinant viruses presented in this study is clearly insufficient. No genome sequence analysis was reported. Growth kinetics of the different viruses showing viral titers (pfu/ml) at different times need to be added instead of measuring an inhibitory effect of the deletion (fig 6B).”

Response:

We agree that “inhibitory effect” might have been a less precise wording for Figure 6B. The experiment has been repeated to include a comparison of the 3 different recombinant viruses in both WT and ATG5 KO Vero B4 cells at the two different time points, assessing genome equivalents as well as plaque forming units (Fig. S4E,F). In WT Vero B4 cells, recombinant MERS-CoV and MERS-del4b grow comparably at 24 hours post infection, whereas the MERS-del5 virus shows approximately 0.5 log₁₀ less efficient growth. At 48 hours, we further confirmed that the two mutants MERS-del4b and MERS-del5 virus grow up to 1 log₁₀ (10-fold) less efficient than the recombinant MERS-CoV in Vero B4 cells using an MOI of 0.001.

New figure S4E,F:

Legend to figure S4E,F: Deletion of ATG5 in the host cells VeroB4 does not abolish the difference in replication between mutant and wt MERS-CoV. VeroB4 wt or ATG5 KO cells were infected with wt or mutant recombinant MERS-CoV (rEMC) (MOI = 0.001) and genome copies (E) as well as plaque forming units (F) (PFU) were determined at 24 and 48 h p.i.. Unprocessed data are presented.

Comment Page 15 and Figure 6 panels A and B, continued:

"More importantly, the authors did not report that in two previous independent studies (ref 44 of this article and Almazan et al, Mbio, 2013), construction of equivalent mutant MERS-CoV (no expression of orf5 or orf4ab) led to different results. Indeed, recombinant viruses lacking orf5 expression replicated to high titers (44). Similarly, Almazan et al observed that virus titers, cpe and plaque morphology of MERS-delta5 are similar to that of parental virus. In the case of MERS-delta4ab the viral titer was only 10 fold lower at 72hpi. How the authors explain this discrepancy? This can be related to an improper way to quantify viral production. Usually viral titers are determined by plaque assay (pfu/ml) in growth kinetics at different times of infection. In this study, they use RT-PCR in supernatant to detect viral genome but this experiment does not measure viral infectivity."

Response:

As shown above, we reproduced the experiment independently and now include the PFU/ml and the GE/ml data in the manuscript (Fig. S4E,F). The discrepancy to the previously published data by Almazan and colleagues may be explained by the different ways of introducing the deletions into the infectious MERS-CoV cDNA clone. In addition, we used a different cell line and a very low MOI which allows to determine minor growth differences. We included the following sentences into the manuscript (end of page 24):

..." Deletion of p4b or p5 resulted in reduced MERS-CoV growth. These results differ from previous observations, which may be related to different cell lines and cDNA clone construction strategies^{72,73}. In addition, we used a very low MOI and a type I IFN-deficient cell line (VeroB4), which allows to determine minor growth differences."...

Comment Figure 6 panel C

The authors concluded that mutant viruses do not control autophagy compared to parental virus. However, no quantification of LC3-II/LC3-I is available and the expression level of LC3-I seems important compared to other WB of the article.

Response:

We now provide quantification of LC3BII/I along with the western blot (Figure S4C,D):

Legend to figure S4C,D: VeroB4 cells were infected with WT or mutant recombinant MERS-CoV (rEMC) (MOI = 0.001) and the indicated proteins were determined at 48 h p.i. The quantification is in reference to control (mock-infection, set to 1).

The reviewer also noted an intriguing point, to which we only can offer a speculation at the moment, namely the apparent effect on the overall levels of LC3B (LC3B-I). We quantified LC3B-I separately, and confirmed a difference, which came out non-significant though. We mention this now in the discussion as follows (starting at the end of page 24): "...The absence of p4b and p5 during MERS-CoV replication may also result in lower LC3B levels compared to MERS-CoV wildtype-infected but also to the mock-infected control cells. This points to a more complex interaction between virus replication and autophagy; p4b and p5 may counteract a response of the host to viral infection, but the functional relevance of their link to autophagy requires further investigation....".

Comment Panel B and C: How is the multiplication of these mutant viruses in autophagy deficient cells? If growth kinetics of mutant viruses are similar to that of parental virus, this would allow the authors to confirm the antiviral role of autophagy on MERS-CoV and to reveal a possible requirement of orf4b or orf5 to counteract it.

Response:

To address this point, we tried gene silencing to inhibit autophagy. However, this turned out not to be successful in the cells we use for the viral replication assays, as transfection and cell growth in combination with viral infection was insufficient (the flux assays with the tandem-labelled LC3B plasmid do not depend on a high transfection efficiency). However, we eventually were able to produce ATG5-/- Vero4B cells. Viral replication in these cells and in WT cells was monitored by PCR assessing genome equivalents and by determining plaque forming units. We found that deletion of ATG5 increased virus growth up to 50-fold on the one hand, but on the other hand does not support the conclusion that p4b or p5 are responsible for this effect. Therefore, we moderated our conclusion to this point. The results are presented in the new figure S4E,F which we copied above in response to another issue of the reviewer.

The section in the results is (begins at the end of page 17):

"Deletion of p4b and p5 led to a decreased accumulation of P62 and LC3B-II/I, suggesting that both proteins contribute to the inhibition of the autophagic flux (Fig. S4C,D). The p4b- and p5-deleted viruses as well as the WT control virus grew to higher levels in ATG5 knockout Vero cells compared to WT cells (Fig. S4E,F). However, the p4b and p5-deleted viruses showed overall an up to 10-fold decreased replication in both WT and ATG5 knockout cells compared to WT virus suggesting a p4b- and p5-dependent attenuation of virus replication that is independent of ATG5-directed autophagy."

Comment Figure 6

The titer of the legend does not reflect the results. No data in this figure demonstrates that SKP2 inhibition reduces viral replication.

Response:

We agree and took care to make sure the titles better reflect the figures. With the new data, figures were rearranged, so figure 6 now largely is integrated into figure 8, but not entirely.

Comment Page 15

"Regarding SKP2 inhibitor, effect on viral replication is impressive, but same remark than above, growth kinetics and determination of viral titers are necessary."

Response:

In the revised manuscript we included a modified version showing the fold change difference between DMSO vs Skp2i-treated cells (now figure 8A). In the supplement (Fig. S5A), we also include the unprocessed data showing the GE/ml. As outlined above (Figure S4E,F) and below (Figures 9B, S9A-C), we showed for several cases that GE/ml can be used as a surrogate for PFU/ml.

New figure 8A and S5A:

Legend to Fig. 8A: The SKP2 inhibitor (SKP2i) efficiently inhibits MERS-CoV replication. VeroB4 cells were infected with MERS-CoV (MOI = 0.001) and treated with SKP2i (10 μ M) or DMSO (Co = control). MERS-CoV GE were determined by real-time RT-PCR at 24 and 48 h p.i., data are presented as fold difference (black bars, SKP2i condition) in comparison to control (grey bars, Co = DMSO condition). Raw data presenting GE/ml is shown in Fig. S5A.

Legend to Figure S5A: SKP2i reduces MERS-CoV replication. VeroB4 wt cells were infected with MERS-CoV (MOI = 0.001) and genome copies were determined at 24 and 48 h p.i.. Unprocessed data are presented from which the fold differences presented in Fig. 8A are derived.

Comment Page 15, continued:

“Moreover, to prove that the compound is acting via autophagy, effect of SMIP004 should be tested in autophagy deficient cells. This kind of approach was used for antibacterial activity of tat-beclin1 for example, in reference 46, Shoji-Kawata et al Nature 2013. ”

Response:

We agree that testing the effect of SMIP004 in autophagy-deficient cells would add useful information. We actually considered performing the mentioned experiment; however, due to reconstruction work of the BSL3 laboratory during the revision, we were not able to perform this experiment in a timely manner. We reasoned that other experiments requested by the reviewers (e.g. providing pfus) needed to be given higher priority in using the very limited BSL3 laboratory availability. Furthermore, we provide strong evidence that a) SMIP004 induces autophagy, b) MERS-CoV benefits from inhibition of autophagy (new experiments in newly constructed Vero ATG5 KO cells), c) SMIP004 inhibits MERS-CoV replication along with reversing the MERS-CoV induced inhibition of autophagic flux, and d) several autophagy-inducing agents also inhibit MERS-CoV replication. We agree though, that

all these compounds might have additional effects and mention this in the discussion, where we adapted our conclusions (page 25): "From the present results, therapeutic induction of autophagy by inhibition of SKP2 emerges as a novel approach. We should note that all the substances tested here have or may have additional targets. While we cannot exclude the possibility that for each of the compounds it is this additional target that is responsible for the antiviral effect, we consider this scenario as less likely; this assessment is also based on the observation of the antiviral effects of compounds targeting BECN1, which we establish here as client of SKP2."

Furthermore, we changed the title of the manuscript to: "SKP2 attenuates autophagy through Beclin1-ubiquitination and its inhibitors combat MERS-Coronavirus infection".

Comment Page 17

"ABT-737 and Tat-Beclin both inhibit MERS-CoV replication and they "leave the levels of BECN1 unchanged". Therefore, the authors cannot conclude just after that "increasing BECN1 is the relevant mechanism". We can make the same remark on page 18, on the mechanism of SKP2 inhibitor by enhancing BECN1 levels. Effects of overexpression of BECN1 on CoV replication should be tested. A siRNA against BECN1 will confirm the involvement of BECN1 and autophagy in CoV replication."

Response:

We agree that the conclusion was not stringent the way it was presented. All we can state is that another and independent mechanism that enhances BECN1's activity in autophagy elicits the same antiviral effect on MERS. This suggests that of the activities potentially elicited by enhancing BECN1 protein levels it is autophagy that is relevant, but it does not prove it. As mentioned above, cotransfection experiments (including overexpression of Beclin1) in combination with viral infection were not successful, unfortunately, in particular using any kind of siRNA (which also would risk to generate off-target effects). We refined the description in the results section (page 20):

"Our data strongly suggest that of the potential mechanisms of SKP2i, it is the increase of BECN1-directed autophagy that reduces MERS-CoV multiplication. If this is correct, other ways of enhancing BECN1's effect on autophagy should elicit similar effects. Therefore, we used two recently introduced tools to enhance BECN1 function in autophagy: the BH3 mimetic ABT-737 and a BECN1-derived peptide^{58,59}. These compounds, in contrast to SKP2i, do not change the levels of BECN1, but increase autophagy by redirecting BECN1 function towards the autophagy pathway, which we confirmed in VeroB4 cells (Fig. S7B-D,G). Enhanced autophagy was indicated by the increase of LC3B-II/I (Fig. S7E,G) and the reduction of P62 (Fig. S7F-G). This was paralleled by inhibition of MERS-CoV replication by up to 60-fold in the case of the BECN1-peptide (TAT-B) (Fig. S7H) and 10-fold in the case of ABT-737 at 48 h p.i. (Fig. S7I). These results are in accordance with the notion that increasing BECN1 acts through enhancing autophagy as the relevant mechanism for the antiviral effects of SKP2 in vitro."

Comment Page 18:

The authors need to report that several of their tested drugs have well-described activity on autophagy, such as niclosamide, valinomycin, salinomycin and gossypol. They only confirmed previous studies. Moreover, these compounds are not only SKP2 inhibitors and specific autophagy inhibitors but have other activities since for example Niclosamide increases mitochondrial fission and valinomycin is a potassium ionophore. Once again, viral replication should be explored using plaque assay and growth kinetics.

Response:

We agree that autophagy induction of these compounds has been shown before. This actually is the reason why the compounds were chosen, as mentioned in the text. The intention was to begin to explore, whether there might be readily available drugs that tap into the pathway presented in the manuscript. We modified the results section, because it could have been misunderstood (page 22):

"These results encouraged us to test other potential inhibitors of SKP2. Autophagy-inducing, FDA-approved drugs and drugs from clinical trials were of particular interest as these could become more readily available for treatment in humans. Of note, valinomycin (VAL), which has been shown to target SARS-CoV in vitro^{50,61}, is also known to act as an SKP2 inhibitor⁴⁹."

As requested, we repeated the experiments to include plaque assays at different time points. Reduction of MERS-CoV growth is now presented as fold change (log10 scale) of both genome equivalents and plaque forming units (new Figures 9B, S9A-C). Raw data (absolute quantification of viral RNA as "genome equivalents, GE" and infectious particles as "plaque forming units, PFU", all presented now in the new Figures S9A,C) of this experiment showed that the difference between GE and PFU is 1000-fold irrespective of the drug used. In other words, whether we used real-time RT-PCR or the pfu assay did not matter, because we obtained the same pattern among all drugs. Therefore, the determination of EC50 values of VAL, NIC and SKP2i was performed only by real-time RT-PCR.

New figure panel 9B:

Legend to Figure 9B: VeroB4 cells were infected with MERS-CoV (MOI = 0.001), treated with the indicated drugs, and MERS-CoV genome copies were determined by

RT-PCR 24 h and 48 h p.i.; data are presented as (\log_{10}) fold difference in relation to the DMSO control. Unprocessed data and pfu results are presented in Fig. S9A-C.

New figure panels S9A-C:

Legend to Figure 9A-C: VeroB4 cells were infected with MERS-CoV (MOI = 0.001), treated with the indicated drugs, and MERS-CoV genome copies (A) as well as PFUs (B,C) were determined 24 h and 48 h p.i.; data represent one of two independent experiments showing the mean values of biological triplicates and are displayed as \log_{10} GE/mL (A) (fold difference in Fig. 9B) and fold difference of the pfus (B) derived from the data presented in panel C.

Comment Figure 7 panels B and C

"The unusable way to determine the antiviral activity does not valorize the impressive results. Even if all these drugs inhibit SKP2, they have other activities. Therefore, the title needs to be modified unless impact of SKP2 is tested by gene silencing."

Response:

Following the reviewer's suggestion, we changed the title of Figure 9 (~previous figure 7) to "Test of potential SKP2 inhibitors for their effects on autophagy and viral

replication”, because silencing of SKP2 by siRNA did not work and appears to be toxic for the cells, unfortunately. We copied the new figures 9B and S9A-C (the experiments of the previous figure 7B have been repeated and expanded to include pfus) in our response to the comment before.

In addition, we changed the title of the manuscript to: “SKP2 attenuates autophagy through Beclin1-ubiquitination and its inhibitors combat MERS-Coronavirus infection”.

Comment Discussion page 21 line 5

“Same remark than above, unless they are testing SKP2 knock down, the authors cannot conclude on impact of SKP2 inhibition on viral multiplication.

The authors should report in the discussion that autophagy has no antiviral effect on MHV, another betacoronavirus.”

Response:

We agree that the conclusion only goes as far as the specificity of the compounds. As detailed above, we modified the title and text accordingly. MHV is now mentioned in the introduction (copied in response to an earlier point above).

Concluding major comment

In conclusion regarding the antiviral activities the authors clearly identified components able to block MERS-CoV replication but the mechanism by SKP2 inhibition is not so far demonstrated.

Response:

As detailed in the response to the specific points above, we modified the text to moderate this specific claim.

Minor remarks:

Comment Figure 1

“(C-F) FKP51.... Transfected with vector control... or FKP51” add (Co) after control”

Response:

We modified the figure panels as requested.

Comment Figure 3 panel I

“Information regarding \$\$ is missing in the legend. Unfortunate that only LLP degradation was tested with siRNA against SKP2 and no other readout for autophagy such as LC3 or p62.”

Response:

We added the missing information to the figure legend (former panel I of figure 3 is now panel M of figure 4). Additional effects observed with the siRNA were the decrease of Beclin1 ubiquitination (Fig. 2I), the increase in Beclin1 levels and stability (Figs. 2G, J-M, S1C), the increase in LC3BII/I (Figs. 2G, S1C) the increase of ATG14 oligomerisation (Fig. S2H,I), the restoration of autophagic flux in MERS-CoV infected cells (Fig. S7J), and the decrease of p62 (new Fig. S1G).

New Fig. S1G:

Legend to Figure S1G: Western blot and quantification of P62 upon downregulation of SKP2 by siRNA in HEK293 cells. All graphs (showing the means \pm SEM are representative of three independent experiments.

Comment Page 10

“Targeting SKP2 INCREASED protein degradation”

Response:

We apologize for this error, which must have been confusing. The sentence is now corrected.

Comment Page 11

“we aimed to test if blocking the SKP2 dependent autophagy inhibition influences CoV replication”. To improve the understanding of this sentence, it is better to say “if stimulation of autophagy by SKP2 inhibitor influences MERS-CoV replication. Add MERS instead of just CoV because, it was tested only for this coronavirus.”

Response:

We agree with the reviewer and thank for this kind suggestion. This section, marking the transition to the viral investigations, is now completely revised, and this specific sentence disappeared. The transition now reads:

"Results up to this point suggested that activity of BECN1 and thereby autophagy is limited by SKP2, which in turn is activated by AKT1 (⁴⁶ and Fig. 3/4). Since MERS-CoV has been shown to enhance phosphorylation of AKT1²⁵, it might reduce autophagy."

Comment Page 13-15

"Several mistakes in the labelling of panels for figure S3

Page 13 S3DE is S3EF; S3F is in fact S3G, S3G is S3H

page 15 S3H is S3I I is J and J is K

Moreover in Figure S3K what is the condition "Co"? same question for Figure S5 panel I. where are the white histograms?"

Response:

We double-checked the panel labels in all figures, and apologies for the inconvenience the errors had caused. In S5I (now S7J), there are no white histograms, because all conditions are with virus (and thus grey according to the figure key, which is placed at this last panel of the figure).

Legend to figure S7J: SKP2i restores autophagic flux in MERS-CoV infected cells. VeroB4 cells were infected with MERS-CoV (MOI = 0.001), treated with SKP2i for 48 h, and incubated with bafilomycin A1 (BafA1, 0.1 μM) for 2 h before samples were taken at 48 h p.i.. The ratios of LC3B-II/I were determined by western blotting and quantified.

Comment pages 2 4 15 17 20 22

"The word "propagation" is used several times in the text, but corresponds in fact to viral cell-to-cell spread and is not tested here. It should be replaced by viral multiplication or production."

Response:

The revised manuscript includes the detection of infectious units using virus plaque assays. As we transferred cell-free virus-containing supernatants to non-infected cells to determine the amount of infectious units, we think that now it would be appropriate to use word virus propagation. Nevertheless, we modified our wording.

Final comment

“Finally the title does not totally reflect data presented in the article, because impact of SKP2 on viral replication has been explored only using compounds which have other activities than just inhibiting SKP2.”

Response:

We changed the title to "SKP2 attenuates autophagy through Beclin1-ubiquitination and its inhibitors combat MERS-Coronavirus infection".

Reviewer #3 (Remarks to the Author):

Autophagy is a cellular homeostatic pathway that can mediate degradation of obsolete organelles, misfolded proteins and other cargo like viral components. Being a destructive process, autophagy has to be tightly controlled. Indeed, intricate signalling pathways regulate both induction and flux rates of autophagy. Recent reports have highlighted the impact of autophagy on many cellular processes and its role as an anti-viral mechanism.

Gassen et al had previously identified FKBP51 as a beclin-binding protein, which mediates induction of autophagy (Gassen et al, 2014&2015). Continuing their work on autophagy, they elaborate on the mechanism. FKBP51 is assembling an autophagy-promoting complex, which deactivates/dephosphorylates the E3 ligase SKP2. Inactivated SKP2 no longer mediates K48-linked polyubiquitination of beclin-1 thereby stabilizing its protein levels and promoting autophagy. Furthermore, the authors characterise the interplay between autophagy and MERS-CoV. While infection inhibits autophagic flux via several viral proteins, exogenous activation of autophagy by pharmacological inhibition of SKP2 combats MERS-CoV infection. Experiments characterising autophagy are conducted in a convincing way and the data supports the claims of the authors. The paper by Gassen et al features an elegant exploration of the complicated mechanism of FKBP51 and links SKP2 to beclin-1. They manage to assemble puzzle pieces from previously published studies about the regulation of SKP2 (e.g. Rodier et al, 2008 and others) complemented by their own data to suggest assembly of a multi-protein complex that regulates stability of beclin-1. Finally, this study provides novel data on how to combat viral infections (here MERS-CoV) using insights gained from understanding the molecular mechanism of autophagy induction and virus-autophagy interplay. However, occasionally it feels like there are two (interesting) separate stories existing side-by-side (Mechanism of FKBP51/SKP2, Interplay between autophagy and MERS-CoV), which need to be tied together more tightly.

A few specific issues need to be addressed as well:

Major:

Comment 1:

"Treatment with drugs can have multiple effects on viral replication. Therefore, even though all used drugs presumably target a similar pathway, it has to be made sure that autophagy is responsible for viral attenuation. The authors should conduct experiments in which autophagy is induced (e.g. by mentioned drugs), but the effects of this induction are blocked. For example, can the growth attenuation be rescued if autophagic turnover is blocked by chloroquine, or LC3B conjugation inhibited by ATG5 KO/KD? Furthermore, the chosen drugs used to inhibit SKP2 (even though it is backed up by literature) need direct evidence that SKP2-dependent beclin-1 ubiquitination is reduced."

Response:

We tried to use chloroquine, but this turned out to be toxic for the cells. Furthermore, ATG5 KD (or any other transfection experiment, in particular siRNA transfection) was not successful in combination with viral infection. Thus, we constructed ATG5 KO in VeroB4 cells and found that MERS-CoV replicates better in these cells. As mentioned in our response to reviewer 2, we agree that testing the effect of SMIP004 in autophagy-deficient cells would be useful. We actually considered performing the mentioned experiment; however, due to reconstruction work of the BSL3 laboratory during the revision, we were not able to perform this experiment in a timely manner. We reasoned that other experiments requested by the reviewers (e.g. providing pfus) needed to be given higher priority in using the very limited BSL3 laboratory availability. Furthermore, we provide strong evidence that a) SMIP004 induces autophagy, b) MERS-CoV benefits from inhibition of autophagy (new experiments in newly constructed Vero ATG5 KO cells), c) SMIP004 inhibits MERS-CoV replication along with reversing the MERS-CoV induced inhibition of autophagic flux, and d) several autophagy-inducing agents also inhibit MERS-CoV replication. We agree though, that all these compounds might have additional effects and mention this in the discussion, where we adapted our conclusions (page 25): "From the present results, therapeutic induction of autophagy by inhibition of SKP2 emerges as a novel approach. We should note that all the substances tested here have or may have additional targets. While we cannot exclude the possibility that for each of the compounds it is this additional target that is responsible for the antiviral effect, we consider this scenario as less likely; this assessment is also based on the observation of the antiviral effects of compounds targeting BECN1, which we establish here as client of SKP2."

In line with these considerations, we changed the title of the manuscript to "SKP2 attenuates autophagy through Beclin1-ubiquitination and its inhibitors combat MERS-Coronavirus infection". This title leaves open which exact molecular pathway might be responsible for the effect of these inhibitors.

Comment 2.

"The interesting data on MERS-CoV prompts the question, how do other viruses, which are known to be sensitive towards autophagy, react towards inhibition of SKP2. Experiments exploring the effect of the drugs on e.g. Sindbis Virus or HIV-1 replication should clarify that issue and would complement the MERS-CoV data and make it less focused on treating a single virus with a system potentially applicable for other viral pathogens."

Response:

This is an interesting question. Autophagy was previously shown to protect from Sindbis virus infection by a p62-dependent mechanism (Orvedahl et al. 2010, CHM). We performed a new set of experiments with Sindbis virus; Sindbis is a fast replicating virus so that the experimental conditions had to be adapted by using a lower MOI (0.0001 instead of 0.001) and including an earlier time point (16 hours).

Sindbis virus also turned out to be somewhat sensitive towards SKP2 inhibition, even though much less pronounced than MERS-CoV. This may be explained by a differential interaction of both viruses with the autophagic flux. The Sindbis data are now included in the manuscript as Figure S5B,C:

Legend to panels (B,C): VeroB4 wt cells were infected Sindbis Virus (SiV) (MOI = 0.0001), treated with 10 μ M SKP2i (SMP004) and pfus were determined at 16 and 24 h p.i.. Data are presented as fold difference in comparison to DMSO vehicle control (co) treated cells (B), derived from the unprocessed data presented in (C).

The discussion now reads as follows (page 25):

"It remains to be investigated whether inhibiting SKP2 will affect all autophagy-sensitive viruses. SINV, which was not overtly influenced by ATG5 knock-out despite indications of degradation of viral components through autophagy⁷⁷, reacted modestly to SKP2i. This may relate to autophagy functioning in the absence of ATG5²."

[redacted]

Comment 3:

"Fig. 2C: To show that SKP2 is the E3 ligase which ubiquitinates beclin-1, the authors should include a SKP2 E3-ligase null mutant and monitor beclin-1 levels/degradation. The effect observed in Fig. 2C-F should be rescued by MG132 treatment, to prove its proteasome dependent degradation."

Response:

We observed that SKP2 null MEFs grow extremely slow. Therefore, we used siRNA against SKP2 as before (the efficiency is documented in Figs 2G/S1C). We performed all the requested experiments and found that siRNA targeting SKP2 increased the stability of Beclin1 and ectopic overexpression of SKP2 had no effect in the presence of MG132, proving a proteasomal-dependent effect. In addition, to more thoroughly address this question we also assessed ubiquitination of Beclin1 in combination with siRNA against SKP2. These data are now included in Figure 2C-F,I-M (with further modifications and conditions on the request of the other reviewers):

Legend to Figure 2C,D: SKP2 affects BECN1 stability. HEK293 cells were transfected with plasmids expressing SKP2 or TRAF6 and exposed to MG132 (10 μ M, 2 h) where indicated and to cycloheximide (CHX, 30 μ g/mL) after 72 h for the durations indicated to monitor the decay of BECN1.

Legend to Figure 2E,F: The conditions of C,D were also used in the pulse-chase assay, performed as in Fig. 1E,F, to determine BECN1 stability.

Legend to Figure 2I: Knock-down of SKP2 by siRNA decreases BECN1 ubiquitination (assay as in Fig. 1B). Representative blots are shown.

Legend to Figure 2J-M: The stability assays of the panels C-F were performed to determine the effect of SKP2-targeting siRNA on BECN1 stability. All graphs (showing the means \pm SEM) and all western blots are representative of three independent experiments.

Comment 4.

"Fig. 3: A few additional details have to be clarified: Is Beclin-1 K402R resistant towards degradation induced by either PHLPPi treatment or SKP2 overexpression? Since AKT1 is involved in the cascade, does pharmacological inhibition of AKT1 lead to stabilization of beclin-1?"

Response:

We compared the stability of Beclin1 WT and the K402R mutant in the cycloheximide assay, including the PHLPP inhibitor. We found that the K402R exhibited greater stability in comparison to WT Beclin1, and further resisted degradation induced by PHLPPi. We also tested the effect of two different Akt1 inhibitors on Beclin1 stability. Both inhibitors increased the stability of Beclin1. These data are now presented as figure 4G-J:

Legend to Figure 4G-J: The cycloheximide protein stability assay was performed (as in Fig. 1C,D) to evaluate the effect of PHLPPi on K402-BECN1 in comparison to wt BECN1 (G,H) and to test the effects of AKT1 inhibitors (AktiX and MK2206) on wt BECN1 (I,J).

Minor comment 1:

The abstract needs rephrasing to emphasise the achievements of this paper in a more cohesive and appealing way. The current abstract undersells the story.

Response:

We thank the reviewer for the kind comment. We tried to emphasize the achievements while avoiding overstatements. Combined with the other reviewers' comments, and in line with this comment, we tried to make sure that the core of our findings comes across, which is a novel pathway for the induction of autophagy. The potential usefulness of compounds addressing this pathway in fighting MERS-CoV infection serves more as an example of possible applications. Whether or not MERS-CoV has evolved mechanisms to tone down autophagy probably is less relevant in this context, but it is a question that naturally comes up and this is why we explored this also. The abstract now focusses more on the novel pathway leading to autophagy induction:

"Beclin1 (BECN1) is a key regulator of autophagy. We here identified S-phase kinase-associated protein 2 (SKP2) as E3 ligase that executes lysine-48-linked poly-ubiquitination of BECN1, thus promoting its proteasomal degradation. FK506 binding protein 51 associates with and stabilizes BECN1 by scaffolding the assembly of a hetero-complex involving PHLPP, AKT1, SKP2 and BECN1 that limits phosphorylation and thus activity of SKP2. Genetic or pharmacological inhibition of SKP2 decreases BECN1 ubiquitination, decreases BECN1 degradation and enhances autophagic flux. Middle East respiratory syndrome coronavirus (MERS-CoV) is an emerging and lethal human respiratory virus for which no specific treatment is available. We found that MERS-CoV multiplication results in reduced BECN1 levels and blocks the fusion of autophagosomes and lysosomes. Inhibitors of SKP2 not only enhanced autophagy but also reduced the replication of MERS-CoV up to 28,000-fold. The SKP2-BECN1 link constitutes a novel target for host-directed antiviral drugs and possibly other autophagy-sensitive conditions."

Minor comment 2:

The title of the paper could be shortened to sound less convoluted.

Response:

We changed the title to: "SKP2 attenuates autophagy through Beclin1-ubiquitination and its inhibitors combat MERS-Coronavirus infection". We tried to simplify the title by constructing it along the two parts (novel mechanism of autophagy regulation and potential application to treatment). As we describe a novel mechanism regarding the SKP2-dependent control of autophagy and additionally show that the induction of autophagy by SKP2 inhibitors has a pronounced effect on MERS-CoV infection we would like to keep both subjects in the title.

Minor comment 3:

Writing of the results section (first part) might need improvement to make it more accessible. Furthermore, mechanistic details could be presented more cohesively to focus on the achievements of the paper.

Response:

Thank you for the kind comment. We rewrote the results section along the lines laid out in the response to minor comment 2.

Minor comment 4:

“Please add size markers to all the western blots.”

Response:

We added size markers to all western blots of figure 1. In our opinion, this makes the figure look very busy, so we suggest to refrain from this in the other figures.

Minor comment 5:

“Fig. 1 B: Stabilization of beclin-1 levels by MG132 has to be compared to NH₄Cl treatment, to rule out turnover by autophagy.”

Response:

We included NH₄Cl in this assay. The results confirm proteasomal degradation of Beclin 1. These data are now presented as Fig. S1B:

Legend to Figure S1B: BECN1 is subject to proteasomal degradation. HEK293 cells were transfected with Ubiquitin-HA expressing plasmid and treated with the proteasome inhibitor MG132 (10 μM, 2 h) in combination with NH₄Cl (10mM) as indicated. BECN1 was immunoprecipitated from whole cell extracts and probed for ubiquitination by western blotting⁸³.

Minor Comment 6:

"Fig. 2A,B: It would be interesting to see the whole Western Blots (especially for USP18 and USP36, including the input)."

Response:

We include whole lane western for USP18 and USP36 below. We will make the other full lane blots available if deemed necessary by the reviewers and/or editors.

(Fig. 2A)

We added a note in the methods section (bottom of page 28 of the supplement): "Transfection of the constructs FLAG-HA-USP18 and FLAG-HA-USP36 each led to the appearance of another band at higher molecular weight in western blots. Irrespective of the exact nature of these bands, specific co-precipitation with FKBP51 was only observable for the bands at the predicted molecular weight (Fig. 1A); these were absent in coprecipitations of BECN1 (Fig. 1B). Figures 1A,B display the bands at the predicted molecular weight."

Minor comment 7:

"Fig. 2C,D: Since SKP2 is regulated by phosphorylation. It would be curious to see how phosphomimetic and phospho-null mutants of SKP2 would impact beclin-1 stability."

Response:

We performed cycloheximide protein stability assays for WT Beclin1 and compared the effect of co-expressed WT SKP2 and the mutants S72D/S75D and S72A (the mutants were described in and obtained from the corresponding author of Nat Cell Biol. 2009 11:397-408). While the phosphomimetic SKP2 S72D/S75D mutant destabilized Beclin1 like WT SKP2, the phospho-null mutant S72A had no effect. These data are now included as figure 4B,C:

Legend to Figure 4B,C: BECN1 stability was assessed by the cycloheximide assay (as in Fig.1C,D) comparing wt SKP2 with the phospho-null mutant S72A and the phosphomimetic mutant S72D/S75D.

Minor comment 8:

"Fig. 3 I: Please show one exemplary western blot of a protein degraded by autophagy."

Response:

We include data on p62 as example of a protein that is degraded by autophagy. These data are now included in Figure S1G.

New Fig. S1G:

Legend to Figure S1G: Western blot and quantification of P62 upon downregulation of SKP2 by siRNA in HEK293 cells. All graphs (showing the means \pm SEM are representative of three independent experiments.

Minor comment 9:

“Fig. 4 E: Please add a quantification for the western blots as shown in the figure.”

Response:

The quantification is shown in figure S2E-G (referring to Fig. 5E, former figure 4E).

Legend to figure S2E-G: VeroB4 cells were treated with the indicated inhibitors (C1 (3.3 μ M), SMIP004 (10 μ M), SMER3 (5 μ M)) and the indicated protein levels were determined (mean + SEM of three independent experiments; quantification to the western blots in Fig. 5E).

Minor comment 10:

“Fig. 5D and E: Both figure panels could be moved to the supplements, as the effect observed is only marginal and other data presented is stronger.”

Response:

Figure 5D is now figure S3D. We agree the effect shown in former figure 5E is quite small. It is hard to predict though, to what degree SNARE protein associations have to change to produce a significant effect on vesicle fusion; furthermore, but less important, former panel E of figure 5 could nicely be accommodated into the new figure 6. Thus, we kept 5E in the main figures (now figure 6H).

Minor comment 11:

“Fig. 6A: While interesting, the sketch could be moved to the supplements.”

Response:

We completely agree and moved the panel to the supplement (Figure S4A).

Minor comment 12:

"Fig. 6G: Could be moved to the supplements"

Response:

Some reasoning as above for Fig. 5E. Former figure 6G is now figure 8F.

Minor comment 13:

"Fig. 6E: While manual counting is certainly being used a lot, these assays could be alternatively quantified using flow cytometry to get data, which might be less biased by manual observation."

Response:

We agree that FACS would be a useful method for this type of assay. We would like to point out that counting was performed by a scientist blind to the conditions. This is now mentioned in the methods description. The effect size is considerable, so it appeared to us that the quantification method chosen is useful. In addition, there is no FACS in the BSL3 laboratory we are using. We were not allowed to export samples from the BSL3 to perform FACS analysis in an external core facility.

Minor comment 14:

"Fig. 7 A,B: Is there a correlation between autophagy induction and MERS-CoV inhibition?"

Response:

In general, the compounds that are more effective in inducing autophagy are also more effective in MERS-CoV inhibition (now Figure 9A,B, expanded by S9A-C). The exemption is the data point of pyrvinium pamoate at 48h p.i. which shows considerable inhibition of MERS-CoV, but no induction of autophagy.

Minor comment 15:

"Fig. 7 B: For the viral inhibition assays, it would be beneficial to show in the supplements the actual copy numbers of viral genome. To get an idea of the magnitude of the effect viral titres for one assay have to be determined showing the effect of autophagy induction on MERS-CoV."

Response:

In response to the comments of reviewer 1 we repeated the experiment to also determine plaque forming units (pfus). Thus, figure 9B (modified from former 7B) is now amended by figure S9A-C. The figures combined show the fold difference as well as the unprocessed data of both assays (determining pfus and genome equivalents per ml).

New figure panel 9B:

Legend to Figure 9B: VeroB4 cells were infected with MERS-CoV (MOI = 0.001), treated with the indicated drugs, and MERS-CoV genome copies were determined by RT-PCR 24 h and 48 h p.i.; data are presented as (log₁₀) fold difference in relation to the DMSO control. Unprocessed data and pfu results are presented in Fig. S9A-C.

New figure panels S9A-C:

Legend to Figure 9A-C: VeroB4 cells were infected with MERS-CoV (MOI = 0.001), treated with the indicated drugs, and MERS-CoV genome copies (A) as well as PFUs (B,C) were determined 24 h and 48 h p.i.; data represent one of two independent

experiments showing the mean values of biological triplicates and are displayed as \log_{10} GE/mL (A) (fold difference in Fig. 9B) and fold difference of the pfus (B) derived from the data presented in panel C.

Minor comment 16:

"Fig. 7C: Is it fold inhibition?"

Response:

Data in former figure 7C (now figure 9C) were presented as fold inhibition in a \log_{10} scale. The figure is now reformatted to display "% virus growth" on the y-axis (\log_{10} scale):

C

Legend to figure 9C: Concentration-dependent inhibition of MERS-CoV replication by the SKP2-inhibitor (SKP2i), valinomycin (VAL) and niclosamide (NIC). VeroB4 cells were infected with MERS-CoV (MOI = 0.001) and treated with drug. MERS-CoV genome copies were determined by real-time RT-PCR at 48 h p.i., data present the percentage of remaining MERS-CoV genomes in comparison to vehicle treatment.

Minor comment 17:

"Fig. 7D: Might be better as a graphical abstract, not as a figure subpanel."

Response:

We like the idea of a graphical abstract. It looks like Nature Communications does not have graphical abstracts; we will check with the editors.

Minor comment 18:

“Page 5: Please add a short sentence with the reasoning why only ‘SKP2, USP18 and USP36 have the potential to influence K48-linked’ etc.”

Response:

*The rationale at the beginning of our study was to focus on USPs and E3 ligases that were found in an interaction assay with FKBP51, the results of which were kindly made available to us by Susan Lindquist. TRAF6 was included as a control, because it is an E3 ligase and FKBP51 interactor, but does not perform K48-linked polyubiquitination (K63 instead, Sci. Signal. **3**, ra42 (2010)).*

The section now reads (end of page 6): “Based on the results of an FKBP51 mammalian-2-hybrid interaction screen³⁸, we first verified the association of FKBP51 with the USPs 18 and 36, and with the E3 ligase SKP2 by immunoprecipitation in HEK293 cells (Fig. 2A). TRAF6 that is known as an E3 ligase of BECN1³⁹ and interactor of FKBP51⁴⁰ served as a control. Of these FKBP51 inter-actors, SKP2, USP18 and USP36 have the potential to influence K48-linked poly-ubiquitination of BECN1. Co-immunoprecipitations of BECN1 in HEK293 cells revealed association with SKP2 only (Fig. 2B), which has not been described as an E3 ligase of BECN1 before. However, inhibition of SKP2 has been previously linked to induction of autophagy^{41,42}.”

Minor comment 19:

“Page 11: mRFP does not resist degradation in the lysosome. It is merely not inactivated by low pH like GFP.”

Response:

We thank for the hint and changed the text accordingly (Legend to figure S9 and methods).

Minor comment 20:

“The authors should discuss the effects of the different viral proteins of MERS-CoV on autophagy, and possible implications in the replication.”

Response:

We agree and included several changes and rewrote the discussion.

The discussion was changed to (page 25):

“For MERS-CoV replication, it remains to be elucidated whether blocking basal autophagy is mandatory or only auxiliary; nevertheless, the virus encodes at least three proteins that limit autophagy when expressed ectopically, NSP6, p4b and p5. Although the detailed molecular mechanisms are yet to be analysed, MERS-CoV NSP6 may also play a role in inhibiting the expansion of autophagosomes as shown

for other betacoronaviruses⁵³. MERS-CoV p4b has phosphodiesterase function inhibiting the activation of RNase L, which itself can activate autophagy⁶⁹. MERS-CoV p5 is located in the endoplasmic reticulum-Golgi intermediate compartment⁵⁶ and was shown to inhibit IFN-beta induction⁷⁰, which may constitute another link to autophagy⁷¹.

Deletion of p4b or p5 resulted in reduced MERS-CoV growth. These results differ from previous observations, which may be related to different cell lines and cDNA clone construction strategies^{72,73}. In addition, we used a very low MOI and a type I IFN-deficient cell line (VeroB4), which allows to determine minor growth differences. Interestingly, the absence of p4b and p5 during MERS-CoV replication resulted in lower LC3B levels compared to MERS-CoV wildtype-infected but also to the mock-infected control cells. This points to a more complex interaction between virus replication and autophagy; p4b and p5 may counteract a response of the host to viral infection, but the functional relevance of their link to autophagy requires further investigation. Apparently, several CoV proteins collaborate to reprogram the membrane traffic in the cell for DMV formation serving as compartments of viral genome replication and transcription^{20,27,74}. Since autophagy is tightly linked to intracellular membrane traffic and turnover, it appears possible that MERS-CoV blocks autophagy not only to evade degradation but also to improve membrane availability."

Reviewers' Comments:

Reviewer #1:

Remarks to the Author:

The extensive revisions in this manuscript add to the already large datasets that was included in the original version. The amount of data in this new version clearly separates between 2 stories. The authors identify how SKP2 effects BECN1 levels via ubiquitination and how inhibition of SKP2 by either siRNA or inhibitory compounds effect BECN1 and autophagy. The report then presents an entirely separate story, on the role of autophagy and MERS-CoV including live virus and how individual MERS-CoV proteins are shown to effect autophagy flux. It is unclear why these 2 stories are combined into this large of a paper and it diminishes the goal of making a clear and organized manuscript. The first story on SKP2 and Beclin1 are clear and direct while the second story on MERS-CoV and how it effects and is affected by autophagic machinery is still unfinished with data that is discordant with the text descriptions of it.

In response to my initial comments:

1. The addition of the raw fold changes and PFU/ml are helpful in determining the effects of autophagy on MERS-CoV replication however the RNA and PFU levels do not correlate with each other. In the added figures of Figure 9B and the supplementary figures associated with it, the fold changes and raw PFU and genome numbers do not correlate with increases and decreases going in opposite directions. There is also no quantitation of significance in these figures making it difficult to interpret as described in the text.
2. In Figures 7, 8 and 9 and the supplementary data associated with the figures, especially supplementary figure 9, there are no significance values for many of the experiments. Does this suggest that the findings are not significant when not calculated? The text claims significant differences however that is not evident in the calculations of the figures.
3. The newly added TRAF6 control data requested in Figure 2 does not correlate with the graph presented. An initial concern for this figure is that ectopic TRAF6 was not included in the pulse chase experiments in Figure 2E and F. The authors have now added in the data showing TRAF6 ectopically expressed in this experiment. The text states, as well as the graph of the western blots, that TRAF6 ectopically expressed in the cells does not effect Beclin 1 levels. However the western blots associated with Figure 2E clearly show a stabilizing effect on Beclin1 protein levels, which does not correlate with the graph shown in Figure 2F. There does not seem to be a way to connect the western blot data with that graphed in Figure 2F. The data suggests that the ectopic expression of SKP2 shows beclin1 degradation at the same level as expression of TRAF6. This is not what is shown in the western blots. It is unclear how the graph could be made from the data provided in the figure.
4. The addition of Figure S2D in response to the question of whether the use of HBSS and BafA1 are inducing autophagic flux as expected is warranted. However again the quantification of the western blot and the images of the western blot are not the same. The graph suggests that the LC3II/I ratio is the same for HBSS and BafA1 treatment however the western blot does not show that same correlation. It is clear in the western blot provided that the 2 sets of data are not equal. While the western blot does show reduction of LC3II in HBSS treatment suggesting autophagic flux, the data do not correlate between the graph and the blot shown.

Reviewer #2:

Remarks to the Author:

The authors had greatly strengthened the manuscript by performing further assays to corroborate their findings and they moderated their conclusions based on reviewer's suggestions.

I do not have new remarks.

I think that this article is now suitable for publication in Nature communications

Reviewer #3:

Remarks to the Author:

The authors have sufficiently addressed all major concerns during the revisions.

A few minor points however, still need to be addressed:

1. Figure S5B,C: Since the effect observed for SINV is relatively low, please include statistics in the figure to show significance of the tendency observed. In the literature about 1log differences were reported previously for SINV in Autophagy negative cells (e.g. Orvedahl et al, 2010)
2. Changes in the discussion on page 25: I would recommend to be careful with relating the SINV phenotype to ATG5 independent autophagy. Rather, other pathways of the immune system may be influencing replication of SINV.
3. Response to minor comment 9: Fig. S2E-G: Please update the axis labelling, is it a ratio? Was it normalized to GAPDH/actin ?
4. Regardless of the availability of BSL3 space (which I agree can be a limiting factor), I still think that experiments in ATG5 KO cells need to be done, even though all hints and all evidence points towards autophagy and the MERS CoV phenotype being connected. However, this might be beyond the scope of this paper.

Point by point rebuttal to the reviewers' comments

Re-Revision NCOMMS-18-17466-T

Reviewer #1

General comment: "The extensive revisions in this manuscript add to the already large datasets that was included in the original version. The amount of data in this new version clearly separates between 2 stories. The authors identify how SKP2 effects BECN1 levels via ubiquitination and how inhibition of SKP2 by either siRNA or inhibitory compounds effect BECN1 and autophagy. The report then presents an entirely separate story, on the role of autophagy and MERS-CoV including live virus and how individual MERS-CoV proteins are shown to effect autophagy flux. It is unclear why these 2 stories are combined into this large of a paper and it diminishes the goal of making a clear and organized manuscript. The first story on SKP2 and Beclin1 are clear and direct while the second story on MERS-CoV and how it effects and is affected by autophagic machinery is still unfinished with data that is discordant with the text descriptions of it."

Response:

We agree that the manuscript might be lengthy and that separating it into two manuscripts might be an option, in principle. In our opinion, however, it is a clear strength of the manuscript that we identified a new component of the autophagic pathway (SKP2) and showed that its inhibition has an antiviral effect against the newly emerging pathogen MERS-Coronavirus. WHO recently listed MERS-Coronavirus to be a major threat to global public health and there are currently no approved antiviral therapies. We agree that mechanistic details on MERS-CoV intervention of the autophagic flux would be favorable. Given the fact that MERS-CoV is a BSL3 pathogen, investigations into mechanistic details are highly complex and time-consuming. Therefore we think that this kind of studies should be dealt with separately at a later time point. We here provide clear evidence that MERS-CoV blocks the autophagic flux either by directly inhibiting ATG14-dependent vesicle fusion or indirectly by downregulation of BECN1. This is new information to the scientific community and will stimulate research in the area of autophagy and coronaviruses.

In conclusion, we think that the combination of data regarding the MERS-CoV-dependent modulation of autophagy and the potential treatability of this virus by addressing the novel SKP2-autophagy pathway is highly beneficial for the manuscript.

Specific comment 1:

"The addition of the raw fold changes and PFU/ml are helpful in determining the effects of autophagy on MERS-CoV replication however the RNA and PFU levels do not correlate with each other. In the added figures of Figure 9B and the supplementary figures associated with it, the fold changes and raw PFU and genome numbers do not correlate with increases and decreases going in opposite directions. There is also no quantitation of significance in these figures making it difficult to interpret as described in the text."

Response:

The comment concerns figures 9B and S9A, which display the fold difference (9B) and the raw data (S9A) of the genome equivalents, as well as figures S9B and S9C, which display the fold difference (S9B) and the raw data (S9C) of the pfu measurements. We realized that there was a mistake in the column coloring for DMSO in Figure 9B which may have caused confusion. We apologize for this and corrected the colors. The DMSO black column (left) and gray column (right) were switched. In addition, we would like to emphasize that one has to compare all gray columns (24 hours post infection) and all black columns (48 hours post infection) separately. In Figure 9B the DMSO control was set to 1 for both time points and cannot directly be related to Figure S9A which shows the raw data in log scale. As growth differences are difficult to identify on a log scale we decided to show fold-change in relation to the DMSO control in Figure 9B. Since the presentation of the fold differences in Figures 9B and S9B are mathematical derivatives of the raw data displayed in Figures S9A and S9C, respectively, there cannot be a conflict between the raw data and the fold differences, unless an error happened in the calculations. We double-checked this, and added a line to the figures with the raw data (S9A and S9C) that denotes the level of the vehicle treatment condition to facilitate the visual evaluation of differences, which are less obvious on log-scale figures.

We agree that genome equivalents and the pfu per ml differ, with the numbers of the pfu being much lower throughout. This is a known phenomenon for coronaviruses as not all RNA genomes are incorporated into infectious virus particles. During virus replication and egress defective interfering particles are generated and released to the cell supernatants. The defective interfering particles are exclusively detected by real time PCR and not by plaques assay. For MERS-CoV, the discrepancy between genome equivalents and infectious units was already described in our previously published MERS-CoV isolation study (Muth et al. 2015; PMID: 26157150).

The reviewer is completely right by saying that a direct comparison between GE/ml (Figure S9A) and PFU/ml (Figures S9C) shows some minor discrepancies for some of the compounds. This may be explained by interference of the respective compounds with the production of infectious viral particles which may subsequently influence the GE/ml to PFU/ml ratio. Mechanistic details of these minor discrepancies were not addressed and would be beyond the scope of this paper.

Overall, however, as explained in our response to the first round of comments, the GE/ml and PFU/ml pattern is highly similar. We are now providing quantification of significance. In addition, we submit all raw data to these figures.

Specific comment 2:

“In Figures 7, 8 and 9 and the supplementary data associated with the figures, especially supplementary figure 9, there are no significance values for many of the experiments. Does this suggest that the findings are not significant when not

calculated? The text claims significant differences however that is not evident in the calculations of the figures.”

Response:

We are now providing quantification of significance. In addition, we submit all raw data to these figures in a data source file that covers all figures.

Specific comment 3: “The newly added TRAF6 control data requested in Figure 2 does not correlate with the graph presented. An initial concern for this figure is that ectopic TRAF6 was not included in the pulse chase experiments in Figure 2E and F. The authors have now added in the data showing TRAF6 ectopically expressed in this experiment. The text states, as well as the graph of the western blots, that TRAF6 ectopically expressed in the cells does not effect Beclin 1 levels. However the western blots associated with Figure 2E clearly show a stabilizing effect on Beclin1 protein levels, which does not correlate with the graph shown in Figure 2F. There does not seem to be a way to connect the western blot data with that graphed in Figure 2F. The data suggests that the ectopic expression of SKP2 shows beclin1 degradation at the same level as expression of TRAF6. This is not what is shown in the western blots. It is unclear how the graph could be made from the data provided in the figure.”

Response:

It appears to us that the reviewer’s impression derives from a combination of our poor selection of the blot example and maybe screen display. We are now submitting less exposed blots to this figure, along with the quantification data of the individual blots in the data source file.

Specific comment 4: “The addition of Figure S2D in response to the question of whether the use of HBSS and BafA1 are inducing autophagic flux as expected is warranted. However again the quantification of the western blot and the images of the western blot are not the same. The graph suggests that the LC3II/I ratio is the same for HBSS and BafA1 treatment however the western blot does not show that same correlation. It is clear in the western blot provided that the 2 sets of data are not equal. While the western blot does show reduction of LC3II in HBSS treatment suggesting autophagic flux, the data do not correlate between the graph and the blot shown.”

Response:

We agree that the western blot did not fully reflect the average given in the graph, partly due to, as we suspect, the higher abundance of LC3I in comparison to LC3II which makes it difficult to select blots that visualize differences in the two LC3 species equally well for the eye. As before, we are now submitting less exposed blots (new experiment) along with the raw data of the quantification.

Reviewer #2

This reviewer has no new remarks and recommends the article now for publication in Nature communications. We are, of course, pleased about this evaluation and comment.

Reviewer #3

This reviewer sees all major points as sufficiently addressed, with a few minor points remaining, however:

Minor point 1: "Figure S5B,C: Since the effect observed for SINV is relatively low, please include statistics in the figure to show significance of the tendency observed. In the literature about 1log differences were reported previously for SINV in Autophagy negative cells (e.g. Orvedahl et al, 2010)"

Response:

We are now including statistics to figure S5B,C (as well as to other figures in response to concerns of reviewer #1) showing that the reduction differs significantly (32% reduction). As SINV is a fast replicating virus the selected cell line, multiplicity of infection, time points etc influence the size of the effect making a direct comparison with other studies very difficult.

Minor point 2: "Changes in the discussion on page 25: I would recommend to be careful with relating the SINV phenotype to ATG5 independent autophagy. Rather, other pathways of the immune system may be influencing replication of SINV."

Response:

We agree to be more careful and are now mentioning that also other signaling pathways may be influencing the SINV phenotype.

"SINV, which was not overtly influenced by *Atg5* knock-out despite indications of degradation of viral components through autophagy⁷⁷, reacted modestly to SKP2i. This may relate to autophagy functioning in the absence of *ATG5*²."

Was changed to (changes underlined):

"SINV, which was not overtly influenced by *Atg5* knockout despite indications of degradation of viral components through autophagy⁷⁷, reacted modestly to SKP2i. The reason for this is unknown; it may relate to the involvement of other immune pathways influencing SINV, for example the JAK/STAT pathway⁷⁸ or stress granule formation⁷⁹, or to autophagy functioning in the absence of *Atg5*²."

Minor point 3: "Response to minor comment 9: Fig. S2E-G: Please update the axis labelling, is it a ratio? Was it normalized to GAPDH/actin?"

Response:

BECN1 (panel E) and P62 (panel G) were normalized to Actin. Figure S2F already represents a ratio (LC3B-II/I), so normalizing both LC3B-I and LC3B-

II to actin (or any other protein) would not change the numbers. While this is mentioned in the supplementary methods, we realized that we mentioned the Actin antibody, but failed to state that it was used for normalization. We are grateful for the hint and added a respective sentence in the supplement (page 23, last sentence).

Minor point 4: "Regardless of the availability of BSL3 space (which I agree can be a limiting factor), I still think that experiments in ATG5 KO cells need to be done, even though all hints and all evidence points towards autophagy and the MERS CoV phenotype being connected. However, this might be beyond the scope of this paper."

Response:

We agree on the usefulness of this experiment. We further adjusted our wording and mention this experiment for future work, also considering the comments of reviewer 1 (discussion, page 29).

Reviewers' Comments:

Reviewer #1:

Remarks to the Author:

The authors have sufficiently addressed all major concerns during the revisions.